# Linking functional traits and demography to model species-rich communities

Loïc Chalmandrier [1,2,3,4,5✉], Florian Hartig [4], Daniel C. Laughlin[3], Heike Lischke [6], Maximilian Pichler [4], Daniel B. Stouffer [5] & Loïc Pellissier [1,2✉]

It has long been anticipated that relating functional traits to species demography would be a cornerstone for achieving large-scale predictability of ecological systems. If such a relationship existed, species demography could be modeled only by measuring functional traits, transforming our ability to predict states and dynamics of species-rich communities with process-based community models. Here, we introduce a new method that links empirical functional traits with the demographic parameters of a process-based model by calibrating a transfer function through inverse modeling. As a case study, we parameterize a modified Lotka–Volterra model of a high-diversity mountain grassland with static plant community and functional trait data only. The calibrated trait–demography relationships are amenable to ecological interpretation, and lead to species abundances that fit well to the observed community structure. We conclude that our new method offers a general solution to bridge the divide between trait data and process-based models in species-rich ecosystems.

---

[1] Landscape Ecology, Institute of Terrestrial Ecosystems, ETH Zurich, Zürich, Switzerland. [2] Landscape Ecology, Land Change Science, Swiss Federal Research Institute WSL, Birmensdorf, Switzerland. [3] Department of Botany, University of Wyoming, Laramie, WY, USA. [4] Theoretical Ecology, Faculty of Biology and Pre-Clinical Medicine, University of Regensburg, Regensburg, Germany. [5] Centre for Integrative Ecology, University of Canterbury, School of Biological Sciences, Christchurch, Canterbury, New Zealand. [6] Dynamic Macroecology, Land Change Science, Swiss Federal Research Institute WSL, Birmensdorf, Switzerland. ✉email: loic.chalmandrier@biologie.uni-regensburg.de; loic.pellissier@usys.ethz.ch

Trait-based ecology has emerged in the last decades as an important avenue for improving our understanding of community assembly and dynamics[1]. Functional traits are canonically defined as "measurable morphological, physiological or phenological features of species that impact their fitness via their effects on demographic features"[2]. The second part of the definition implies that functional traits should be linked to species' demographic rates and by extension to species ecological niches[3,4]. Yet, while the field has made major progress in collecting, organizing, and analyzing functional trait data for a large number of species from a variety of ecological communities[5], attempts to formally demonstrate broad and consistent links between functional traits and species demographic rates across ecosystems have been less successful[6,7].

The absence of a predictive link from traits to demographic rates not only challenges our ecological understanding, but also poses important practical limitations to our ability to predict community structure and dynamics with process-based community models. Here, by "community model", we refer to any process-based model that predicts community structure and/or dynamics as a consequence of population-level processes such as growth, resource acquisition, mortality, and species interactions[8]. Model processes can be formulated across a range between phenomenological to more mechanistic descriptions[9], but are generally specified by demographic rate parameters that vary across species. By predicting features such as species abundance, community structure, and dynamics over time, ecologists have argued that community models avoid many limitations of correlative models[10–12], and would represent an important step towards predictions of local biodiversity responses to environmental changes[13–15].

A major setback for the agenda of using process-based models for community ecology is that even conceptually simple community models, such as Lotka–Volterra models, are challenging to calibrate. This is because the number of demographic rate parameters of such models increases rapidly with the number of species[13,16,17]. Consequently, the use of process-based community models has been limited to systems with low to moderate complexity and diversity where a direct measurement of these parameters requires less effort, such as annual plant communities[18,19] or laboratory communities[20]; and they have yet to be transferred to natural, species-rich systems.

Here, we propose a new method for calibrating process-based community models via functional traits. Functional traits are far easier to measure across a large number of species than are demographic rate parameters; therefore the ability to parameterize community models via functional trait data would expand their utility considerably. We aim to establish a transfer function that links empirically measured functional traits with the parameters of community models that describe species demography, hereafter called "demographic parameters". The approach assumes that the demographic parameters of the modeled species can be predicted from their functional traits[3]. However, while the idea that form determines function is widely accepted, it would be challenging to predict the nature of this relationship only from a priori assumptions. For example, combinations of functional traits rather than single traits may be necessary to predict demographic parameters[7,21]. Instead, we use one additional data type, empirical community abundance data, to calibrate these trait–demography relationships through an inverse modeling approach[16,22]. In the end, the number of parameters in the transfer function scales only with the number of functional traits and the complexity of the transfer function. This is a critical advantage compared to a direct estimation of the demographic parameters, which scales with the number of species (Fig. 1).

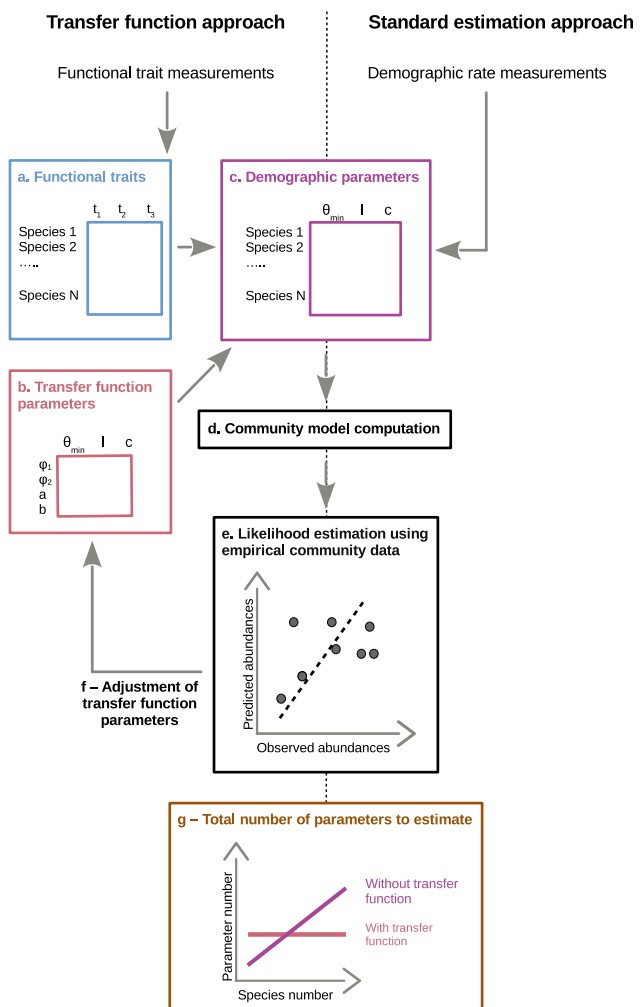

**Fig. 1 Illustration the modeling procedure.** From a data table (**a**) of three empirical functional traits data ($t_1$, $t_2$, $t_3$), we use a "transfer function" (**b**) controlled by a set of transfer function parameters $\{\varphi_{i,1}, \varphi_{i,2}, a_i, b_i\}$. Together they predict demographic parameter values across species (**c**). The community model (**d**) is then run based on the demographic parameter table. The likelihood of empirical community data given the parametrization is estimated using an appropriate likelihood function (**e**). The latter depends on the nature of the community model output and of the observed data (e.g. species abundances, species presence/absence, community diversity etc...). The likelihood value sets the next iteration of the distribution sampler by adjusting the transfer function parameters (**f**).

The proposed method can support the calibration of any process-based community model (e.g. competition models, trophic models[23,24]) to empirical ecosystems as long as a relationship between traits and demographic parameters exists and sufficient functional trait data and community data are available. Here we illustrate the potential of our methodology by calibrating a simple community model with data from 18 communities distributed along a temperature gradient in the French Alps. The communities are composed of 118 mountain grassland plant species characterized by eight functional traits. More specifically, we test the hypothesis that the plant community structure follows the stress-dominance hypothesis: community composition is determined by plant ability to tolerate abiotic stress at the stressful end of the gradient (here at cold temperatures), and by plant ability to withstand competition at the benign end of the stress gradient (here at warm temperatures)[25].

We formulate a community model derived from a Lotka–Volterra competition model that mimics these processes with four demographic parameters for each of the 118 species, three of which could be estimated with static community data. To do so, we use a transfer function to predict those 354 demographic parameters of the model from empirical functional traits through generalized linear functions using a total of only eleven parameters. Second, after the model calibration, we examine the quality of predictions against a set of null models calibrated from randomized trait data to demonstrate that empirical functional trait data bring significant improvement to the calibration. Third, we interpret the calibrated transfer function in the context of the ecological theory regarding the link between species demographic strategy and functional traits in mountain environments[26,27].

## Results

**The community model**. We formulated a simple community model derived from a Lotka–Volterra competition model (Eq. 1) that mimics the processes of the stress-gradient framework: plant biomass growth is modeled as a function of temperature and plant competition.

$$\frac{1}{B_{ij}} \frac{dB_{ij}}{dt} = g_i \times \left( \theta_j - \theta min_i \right) - c_i B_{ij} - l_i \sum_k B_{kj} \qquad (1)$$

$B_{ij}$ represent the biomass of species $i$ in the site $j$ characterized by temperature $\theta_j$. The model includes the following ecological processes:

*Temperature-dependent growth*: $g_i \times (\theta_j - \theta min_i)$ is the intrinsic relative biomass growth rate—incorporating reproduction, mortality, and individual biomass growth—of a species $i$ in a site $j$. It is formulated as a positive linear function of the standardized temperature $\theta_j$ of the site. It is controlled by the parameter $\theta min_i$ that can be interpreted as the minimum temperature above which species $j$ can have a positive relative biomass growth rate; below that threshold species $j$ goes extinct. It is controlled by a slope parameter $g_i$ that can be interpreted as the within-species variability of the growth rate along the temperature gradient.

*Sensitivity to surrounding biomass*: the relative biomass growth rate of a species $i$ in a site $j$ decreases linearly with the total plant biomass of site $j$, $\sum_k B_{kj}$, at a rate determined by demographic parameter $l_i$. Compared to a classical Lotka–Volterra model, this equation simplifies the formulation of biotic interactions by assuming that a given species is equally sensitive to all other competitors and only affected differently by its own biomass.

*Intraspecific competition*: the relative biomass growth rate of a species $i$ in a site $j$ decreases linearly with the biomass of its conspecifics at an additional rate determined by parameter $c_i$. This term is based on the theoretical expectation and the empirical finding that intraspecific competition tends to be superior to interspecific competition[28]. When further accounting for a species' general sensitivity to surrounding biomass, species relative growth rate decreases linearly at an overall rate of $c_i + l_i$ with the biomass of its conspecifics.

Depending on the strength of trade-offs among demographic parameters, the model can return different species abundance patterns along a temperature gradient going from the dominance of a single species along the gradient to strong species turnover[12,29]. The model was implemented as an ordinary differential equation system model (ODE).

**Estimating demographic parameters with a transfer function**. The modeled processes are controlled by four unknown demographic parameters for each of the 118 species: minimum tolerated temperature, within-species variability of the growth rate along the temperature gradient, sensitivity to surrounding biomass, and intraspecific competition. In the absence of temporal data about our studied plant communities, we assumed that they were at equilibrium and could be modeled from the ODE equilibrium. Because of this, within-species variability of the growth rate along the temperature gradient could not be estimated and was thus kept constant across species (see Methods and Supplementary information 1.4). To estimate the remaining 354 demographic parameters, we used generalized linear functions between each demographic parameters and three PCA trait axes that summarized 67% of total trait variance. The eleven parameters of those transfer functions were then calibrated with a Markov-Chain Monte Carlo algorithm (see Methods for more details).

**Model validation**. The calibrated model predicted the expected pattern of species turnover along an environmental gradient (Fig. 2a). Nagelkerke's pseudo $R^2$ value[30] of the calibrated model was 0.590 (median value across the posterior, 95%CI: [0.587, 0.592]) and the deviance information criterion[31] (DIC) was 6476.4. In contrast, the pseudo $R^2$ at the median of the posterior of each of 200 null models with randomized traits was always less than 0.179, and we found a mean difference in DIC of 1844 in favor of the calibrated model (DIC of the null models was distributed within the 95%CI: [7719.4, 9205], against 6476.4 with real functional trait data). For 15 out of 18 plots, the calibrated model performed better than the null models (i.e. for those plots, the pseudo $R^2$ values was superior to 95% of the null pseudo $R^2$, Fig. 2b). In comparison, species distribution models and joint species distribution models similarly capture the abundance of well-sampled (aka dominant) species (see Supplementary information 3) but differ in the modeling of species that were undersampled along transects[32]. Owing to a much larger number of parameters (between 354 and 1062 for the SDM and the jSDMs vs. 11 for our approach), they also fit more closely the transect data (Nagelkerke's pseudo $R^2$: [0.760, 0.961], DIC: [3182.82, 13269]) but have a similar performance to predict plant species presence/absence (AUC: [0.68–0.79] against 0.68 for our approach).

**Trait–demography relationships**. To study the output of the calibrated community model, we assessed Pearson's correlation coefficients among the calibrated demographic parameters, the correlation between demographic parameters and observed functional traits across the posterior distribution. Then we tested if the distribution of calibrated demographic parameters differs among functional groups at the median of the posterior distribution. Functional groups were forbs (76 species), grasses (26 species), legumes (11 species), and shrubs (5 species). We related demographic parameters to the observed species functional trait values rather than to the PCA trait axes used for the calibration to facilitate the ecological interpretation of our results. The posterior distribution of the transfer function parameters is available in the Supplementary Information (Supplementary Fig. 5).

The estimated demographic parameters showed ecologically sensible trade-offs (Fig. 3b): the minimum tolerated temperature (**θmin**) was strongly negatively correlated with the sensitivity to surrounding biomass (**l**) (95%CI: $-0.977 < r(\textbf{θmin}, \textbf{l}) < -0.959$). We predicted demographic strategies[26,33] along an axis characterized by "competitive" species with a weak tolerance to low-temperature stress, but less sensitivity to competition on the one end, and "stress-tolerant" species with the opposite demographic strategy on the other end[26,33].

Those two demographic parameters were correlated with functional traits as predicted by ecological theory[4,27,34]. Plant

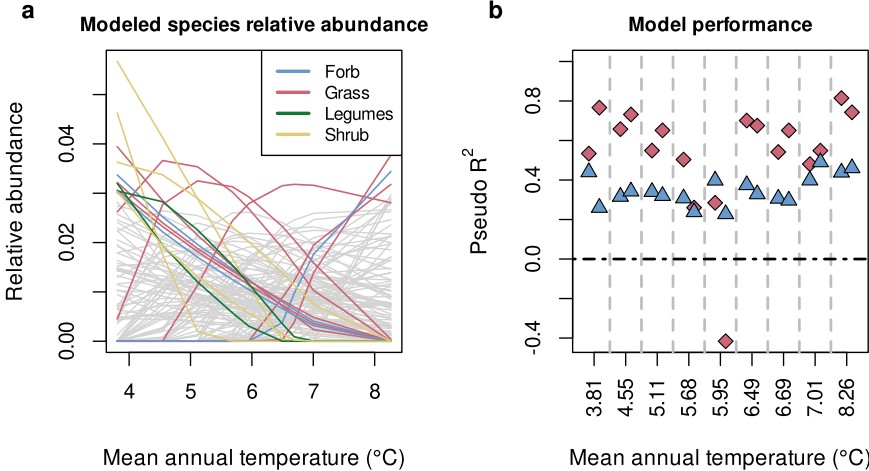

**Fig. 2 Validation of the calibrated community model. a** Modeled species relative abundances along the temperature gradient. The dominant species (able to reach a relative abundance of 0.03 in at least one community) are colored according to their functional group (forb in blue, grass in red, legumes in green and shrub in yellow), subdominant species are in gray. **b** Performance of the community model for each plot across the elevation gradient according to Nagelkerke pseudo $R^2$. Red diamonds indicate the pseudo $R^2$ as predicted by the community model calibrated with the empirical functional traits. Blue triangles indicate the 95% quantile of the distribution of pseudo $R^2$ predicted by the null models. Negative values of the pseudo $R^2$ indicate cases where the likelihood of the model is lower than the likelihood of the hypothesis that all sampled species have equal relative abundance. For each model, pseudo $R^2$ values were calculated at the median of the posterior.

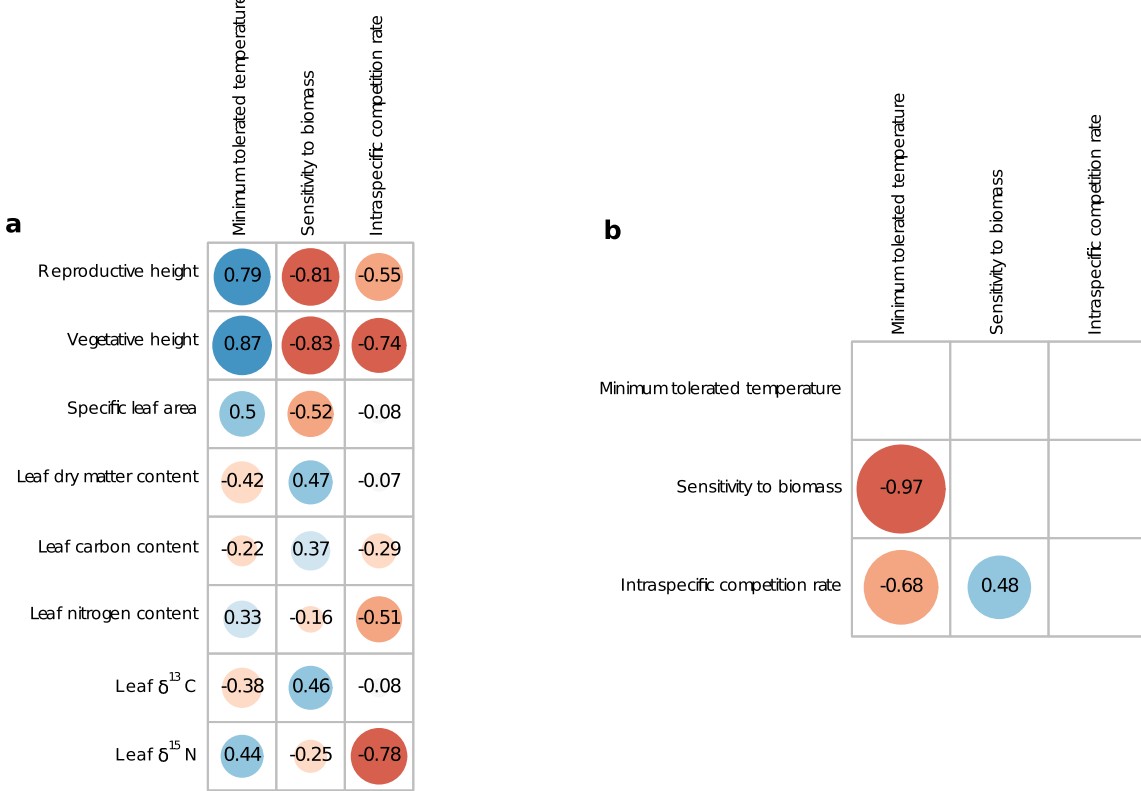

**Fig. 3 Calibrated trait–demography relationships.** Calibrated correlation between demographic parameters and empirical functional traits (**a**) and among demographic parameters (**b**). In each panel, numbers indicate the median of the Pearson's correlation coefficients given by the posterior distribution. Circle size is proportional to the absolute value of the correlation coefficients and its color indicates the value of the correlation coefficients (with red representing a strongly negative correlation value, blue a strongly positive correlation and paler shades small correlations).

species with a tall stature or with tender leaves were associated with a high minimum tolerated temperature and lower sensitivity to surrounding biomass[26,35]: minimum tolerated temperature ($\theta$min) and sensitivity to biomass (l) were strongly correlated with vegetative height (across the posterior distribution, 95% CI,

$0.861 < r(\theta\mathbf{min}) < 0.883$, $-0.843 < r(\mathbf{l}) < -0.806$, Fig. 3a), reproductive height (95% CI, $0.774 < r(\theta\mathbf{min}) < 0.810$; $-0.832 < r(\mathbf{l}) < -0.792$), and moderately correlated with specific leaf area (95% CI, $0.482 < r(\theta\mathbf{min}) < 0.519$; $-0.538 < r(\mathbf{l}) < -0.488$) and LDMC (95% CI, $-0.445 < r(\theta\mathbf{min}) < -0.399$; $0.436 < r(\mathbf{l}) < -0.494$). To an

## a    min. tol. Temperature

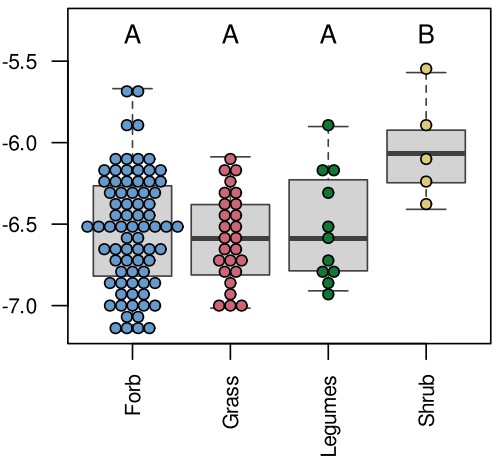

## b    Sensitivity to biomass rate (log)

## c    Intraspecific competition rate (log)

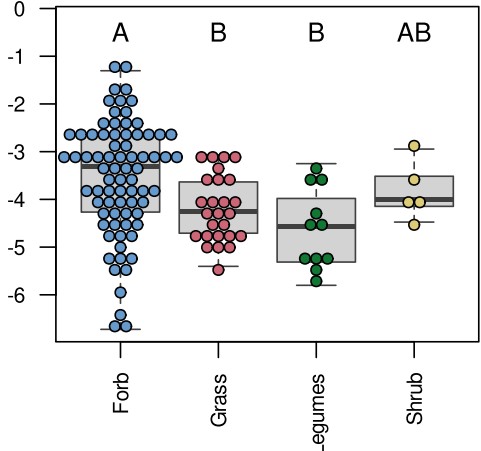

**Fig. 4 Estimated demographic parameters across functional groups.**
Minimum tolerated temperature (**a**), sensitivity to biomass rate (**b**) and intraspecific competition rate (**c**) were estimated from the median of the posterior distribution of the parameters. Functional groups were forbs (76 species), grasses (26 species), legumes (11 species) and shrubs (5 species). Boxplots indicate the median, first and third quartiles of each distribution. Whiskers represent the minimum and maximum values that remain inferior 1.5 times the interquartile range below or above the distribution median. Anova tests indicate that all three demographic parameters were significantly different among functional groups ($\theta_{min}$: $F = 2.742$, $p = 0.047$; **l** $F = 3.719$, $p = 0.013$; **c** $F = 4.669$, $p = 0.004$). Uppercase letters above the boxplots symbolize the results of pairwise t-tests on demographic parameters among functional groups. Distinct letters characterize significantly different demographic parameter distributions among functional groups ($p$ value $\leq 0.05$ after adjustment by Holm's correction). Note that min. tol. Temperature is relative to a mean annual temperature standardized across the study sites and is therefore unit-less.

$\theta_{min}$: Forbs, $t = 2.518$, $p = 0.013$; Grasses, $t = 2.658$, $p = 0.009$; Legumes: $t = 2.712$, $p = 0.008$; **l**: Forbs, $t = -3.204$, $p = 0.002$; Grasses, $t = -3.279$, $p = 0.001$; Legumes: $t = -2.644$, $p = 0.009$). This suggests that shrubs (the latter category includes five alpine low shrub species) were more resistant to low temperature but had lower competitive ability than other species across the temperature gradient.

The intraspecific competition parameters (**c**) were moderately correlated with the minimum tolerated temperature parameters (95%CI: $-0.741 < r(\theta_{min}) < -0.593$) and sensitivity to competition (95%CI: $-0.400 < r(\textbf{l}) < -0.560$). They were further negatively correlated with vegetative height (95% CI, $-0.787 < r(\textbf{c}) < -0.679$, Fig. 3a) and reproductive height (95% CI, $-0.616 < r(\textbf{c}) < -0.481$), showing that tall plant species were associated with higher competitive ability than small species[37]. It was further strongly correlated with leaf $\delta^{15}N$ (95% CI, $-0.801 < r(\textbf{c}) < -0.761$) and, to a lesser extent, leaf nitrogen content (95% CI, $-0.557 < r(\textbf{c}) < -0.463$). Hence, species with a low leaf nitrogen content as well as a low leaf $\delta^{15}N$ ratio (that usually characterizes mycorrhizal dicots[38]) were also associated with higher intraspecific competition. The model parametrization associated a higher intraspecific competition rate with forb species compared to grass and legume species (Fig. 4, Grass: $t = -2.625$, $p = 0.001$; Legumes: $t = -3.101$, $p = 0.002$, Shrub: $t = -0.554$, $p = 0.58$). This association likely reflects the higher competitive ability of grasses and legumes over the other functional groups, which may explain their dominance in these grassland communities.

## Discussion

**General approach.** Combining functional trait data with static community abundances, our approach allowed us to successfully fit a process-based community model to a species-rich ecosystem. Compared to other trait-based modeling methods[39], our approach is not tied to any particular framework of community assembly. It rather establishes, based on empirical data, a plausible link between measured functional traits and the unknown demographic parameters of a community model. To do so, it leverages a fundamental insight of trait-based ecology: species functional attributes vary only across a limited number of dimensions[40], and these should be relatable to species demography. When translating this insight into a transfer function between empirical traits and the demographic parameters of community models, the number of parameters to be calibrated is much reduced, which makes it possible to fit the combined model to static data despite the large number of species. An important

extent, these calibrated relationships match the trait-environment linkages suggested by a fourth corner analysis[36] (see Supplementary information 3.2.2). This suggests that these two demographic rates depend on functional traits that vary strongly along the temperature gradient. Our model further estimated a lower minimum tolerated temperature and higher sensitivity to biomass for shrub species compared to other functional groups (Fig. 4,

point is that our approach does not rely on a priori assumptions about the demographic trade-offs and relevant trait–demography relationships, they rather emerge from the inverse modeling approach.

In our case study, we made the conjecture that species demography follows the stress-dominance hypothesis, and we quantified the support for this theory using the estimated demographic trade-offs. These corresponded to (i) a competition-stress tolerance trade-off that matched closely traits associated with the fast-slow leaf-economics trade-off[34] and (ii) an axis of variation in intraspecific competition intensity that matched traits related to plant size. These relationships relate to existing knowledge about the global spectrum of plant trait variation and its relationship to plants' ecological niche and demography[3,4,41]. While the functional trait–demography relationships calibrated by our approach are unsurprising, they are ecologically sensible and allow us to validate the fit of the transfer function a posteriori. They further suggest that, when moving towards more complex demographic models, a lack of unanimous knowledge about demography-trait relationships may not be an insurmountable obstacle.

**Developing more realistic community models**. Beyond the scope of our particular community model, this new method paves the way to study more complex—and potentially better-performing models—of community assembly and dynamics in diverse ecosystems. We foresee no fundamental difficulties in using transfer functions to calibrate a broader variety of ecological processes. The only requirement to implement new processes is that they should be formulated as a function of attributes linked to species-specific (or individual-specific) functional traits. Some processes, typically biotic interactions, are typically modeled as attributes of two or more species[19] and would need to be reformulated. Several theoretical studies have shown a path forward, for instance by expressing pairwise interaction coefficients from Lotka–Volterra competition models as functions of species' ability to acquire resources[42,43]. Empirical studies have further suggested how pairwise interactions among species can be modeled from their functional traits[6,44], which could inspire new theoretical frameworks that consider other features of biotic interactions such as niche partitioning among species[23], facilitation[45] or trophic interactions[24].

**Towards dynamic community models**. We assumed that the observed plant communities were at equilibrium, a common assumption in spatial community modeling[46]. Furthermore, we chose a globally-stable community model (Supplementary information 1.3), which assumes that, for a given set of demographic parameters values and environmental conditions, a unique stable community structure will emerge. These assumptions make the analysis more practical, but they may not always hold in nature[12,46]. In our case study, assuming community equilibrium led us to fix one demographic rate across species (see Methods), thus limiting our ability to fully model the interspecific variability of species' responses to temperature and biotic stress. While this is arguably reasonable when modeling a relatively small static dataset without temporal data, it would be less adequate to model larger spatio-temporal datasets where it would be more essential, and feasible, to distinguish and calibrate detailed demographic processes.

Theoretical ecology has produced a vast corpus of models that study communities in terms of spatio-temporal dynamics, population structure, stability, and alternative stable states[47,48], with some approaches making explicit references to functional trait theory[49,50]. With adequate community models, our transfer function approach could be used to move away from the equilibrium assumption, and to model spatial and temporal variation of species-rich communities. Besides the important data requirements, however, more complex models would create new analytical challenges as simulation approaches (e.g. ODEs) might not be appropriate to efficiently characterize community dynamics and equilibrium states[51]. In this context of increasing model and data complexity, assuming a priori that species demographic parameters depend on a limited number of functional traits will likely be a critical asset to study the model behaviors and reduce the complexity of its calibration at the onset.

**Comparing alternative modeling frameworks**. Our case study uses environmental data, species abundances, and functional traits to test the stress-dominance hypothesis. An interesting question is how our framework compares with alternative, existing methods. One such approach to analyze static community data is (joint) species distribution models ((j)SDMs), which directly fit a relationship between environmental predictors and species occurrences or abundances[10], without considering species traits or underlying demographic processes. Alternatively, trait-based community analyses, such as the fourth corner method[36], also relate environment variables to species traits on the basis of species occurrence while other trait-based methods go a step further and use trait–environment relationships to predict local species abundances[52,53].

(j)SDMs and the fourth corner method detected, to an extent, the same patterns as the transfer function approach (see Results and Supplementary information 3.2.1). A key difference, however, is that they cannot infer dynamic community parameters, while our method explicitly establishes a link between traits, demography, and abundance. Furthermore, our approach is explicit about the hypotheses underlying the modeled demographic processes. We were directly able to evaluate whether community assembly follows the stress-dominance hypothesis and quantify the support from the data.

A possible drawback of a process-based approach is that it is constrained by both the presumed functional form of the mechanism, and the existence of a statistical link between the processes and the available functional traits. We suspect that this explains why, in our case study, (j)SDMs achieve a closer fit to the data (Supplementary materials 3.2.1) than our approach: (j)SDMs have a much larger number of parameters, and are thus far more flexible, and they do not require a connection between environmental preferences and the available functional traits. We speculate, however, that our approach could be improved by confronting multiple alternative transfer functions and community models. In doing so, it opens new avenues to empirically test and compare alternative process-based models with various degrees of complexity as long as there is a link between the studied demographic processes and the available functional traits. As process-based models imply causal explanations for observed patterns[8,54], this can lead to a promising hypothesis-driven confrontation of community models with different structures.

The key advantage of our approach is that it allows parametrization of process-based models with functional trait data and static community data, two data types that are more easily accessible than labor-intensive demographic measurements. We acknowledge that the proposed transfer function cannot predict the full intricacies of species demographic responses[55], but we believe that it constitutes a reasonable approximation to build dynamic models of species-rich ecosystems in which ecologists are unlikely to ever obtain demographic measurements for every individual species. If the link between demography and

functional traits is reasonably strong, we can draw on all the advantages of functional trait data: they are standardized, reasonably easy to measure across species and environments, and increasingly collected into large open databases[5]. With our approach, the main challenge of developing process-based models shifts from data gaps towards developing appropriate statistical frameworks that transfer functional traits into model parameters. We expect that our approach will promote the use of parameterized process-based community models to explore current theories of community assembly and ultimately better predict the dynamic of biodiversity in a changing world.

## Methods

### Dataset

*Study area.* The study was conducted in the central French Alps (45.12°N, 6.40°E) during the 2012 summer season. Nine sites, each containing two 100 m² quadratic plots, were studied along a continuous elevation gradient (1858–2724 m) on a single south-east facing slope in a cow-grazed pasture. Mean annual air temperature varied between 0.8 °C and 3.8 °C and annual precipitations between 919 and 1186 mm[56]. Subalpine grasslands dominated the bottom of the gradient while sparsely vegetated alpine meadows characterized higher elevations. The nine sites were evenly distributed along the elevation gradient that represents the main local abiotic gradient. Mean annual soil temperature was estimated in each site using soil thermocaptors and varied between 4.55 °C and 8.26 °C. Temperature values were then standardized.

*Community data.* In each square-shaped plot, 101 plant individuals were sampled in July 2012 along two transects that followed its diagonals[57]. 118 species were sampled at least once and on average, 28.0 species were sampled in each plot. The 118 species included 26 grass species (Poales), 11 legumes species (Fabaceae), five shrubs species (Vaccinium, Daphne, and Salix sp.), and 76 forb species (remaining species). This sampling scheme tended to favor the sampling of locally dominant species over low-abundance species[32]. Botanical surveys were further done on the same plots simultaneously to collect more exhaustive data on species occurrences.

*Functional traits.* For each sampled individual, we identified the species and measured eight functional traits using standardized protocols[58]. (i) Reproductive and (ii) vegetative height are associated with plant competitive ability[37]. (iii) Specific leaf area (SLA) is usually correlated positively with plant growth rate and negatively with leaf lifespan[3]. (iv) Leaf dry matter content (LDMC) is related to the average density of leaf tissues and tends to scale negatively with SLA. (v) Leaf nitrogen concentration (LNC) quantifies the allocation of available nitrogen to photosynthetic enzymes in leaf chloroplasts[59]. (vi) Leaf carbon concentration (LCC) represents investment in structural tissues[60]. (vii) Leaf carbon isotopic ratio ($\delta^{13}C$) provides a time-integrated measure of stomatal conductance[61]. (viii) Leaf nitrogen isotopic ratio ($\delta^{15}N$) reflects the isotope signature of nitrogen sources of the plant and thus provides a measure of the plant's nitrogen acquisition strategy[38]. Details about the traits and the measurement protocol are available in the Supplementary information.

*Definition of the species pool's functional trait space.* Trait measurements were averaged by species. All traits except $\delta^{13}C$ and $\delta^{15}N$ were log-transformed to better approach a normal distribution. To optimize the number of parameters used in the transfer function, we ran a Principal Component Analysis (R-package ade4[62]) on the species-trait matrix, retained and scaled the three first orthogonal empirical functional trait axes ($t_1$, $t_2$, and $t_3$) that collectively explain 67.0% trait variance. The three trait PCA axes can be described as follows (Supplementary Table 1 and Supplementary Figs. 1–2):
- $t_1$: this axis represents 29.5 % of the total trait variance. It is negatively correlated with vegetative height, reproductive height, SLA, leaf nitrogen content, foliar $\delta^{15}N$ and positively related to LDMC and foliar $\delta^{13}C$. The analysis of species distribution along this axis indicated that species typical of subalpine grasslands were associated with negative scores while species typical of alpine grasslands (including all five shrub species) were associated with positive scores.
- $t_2$: this axis represents 19.5% of the total variance. It is positively correlated with SLA and negatively related to vegetative height, LDMC, leaf carbon content, foliar $\delta^{15}N$, and foliar $\delta^{13}C$. The analysis of species distribution along this axis indicated that positive scores were associated with forbs species.
- $t_3$: this axis represents 17.9% of the total variance. It is positively related to reproductive height and LDMC and negatively related to leaf nitrogen content and foliar $\delta^{15}N$. The analysis of species distribution along this axis indicated that legumes and shrub species were associated with negative scores while positive scores were associated with grass species.

**Transfer function.** In principle, any mathematical relationship between traits and demographic parameters could be specified as a transfer function. Given a matrix of N empirical functional traits $\mathbf{T} = \{t_{i,n}\}$, where $t_{i,n}$ is the known value of

functional trait $n$ of species $i$, and a matrix of unknown demographic parameters $\mathbf{D} = \{d_{i,m}\}$, where $d_{i,m}$ is the value of demographic parameter $m$ of species $i$, the transfer function specifies the mathematical link between the two. In our case, we used a linear function of the functional traits to specify $\theta\text{min}$ as it can take both positive and negative values across species and a log-linear function for **l** and **c**, which can take only positive values.

We used a hyperspherical parameterization of the regression coefficients of the linear and log-linear functions[63,64]. This formulation defines the relationship between demographic parameters and functional traits with two sets of parameters: a first set that controlled the link between demographic parameters and functional traits, and a second set that controlled the mean and standard deviation of demographic parameters across species.

1—The regression coefficients of the linear expressions linking the species demographic parameters values $d_{i,m}$ to the functional trait values $t_{i,n}$ can be viewed as coordinates lying on a unit hypersphere of dimension N. We express them using the multidimensional extension of the transformation of Cartesian coordinates to polar coordinates. In that scheme, the coefficients are defined by $N-1$ angle parameters $\{\varphi_{m,n}\}$. This parameterization samples efficiently all possible correlations between each demographic parameter and functional trait while keeping constant the mean and standard deviation of the demographic parameters (Supplementary information 1.4.4). For each demographic parameter $m$ of species $i$, we calculate the link scale $E_{i,m}$ that depends on the trait matrix $\mathbf{T}$ containing N traits through $N-2$ parameters $\{\varphi_{m,n}\}$ that vary within the range $[0, \pi]$ and one parameter $\varphi_{i,N-1}$ that varies within the range $[0, 2\pi]$.

$$E_{i,m} = \cos \varphi_{m,1} \times t_{i,1} + \cdots + \prod_{k}^{n-1} \sin \varphi_{m,k} \times \cos \varphi_{m,n} \times t_{i,n} + \cdots + \prod_{k=1}^{N-1} \sin \varphi_{m,k} \times t_{i,N} \quad (2)$$

In the situation, where all traits are orthogonal and follow a standard normal distribution across species, the link scale $E_{i,m}$ also follows a standard normal distribution across species.

2—We proposed two functions to transform the above expression to an appropriate distribution for the demographic parameter $d_{i,m}$ across species. However, any monotonically increasing transformation of $E_{i,m}$ could be employed depending on the data structure. Both expressions depend on additional parameters $a_m$ and $b_m$ that together control the mean and standard deviation of $d_{i,m}$ across species.

1) If $d_{i,m}$ only takes positive values, we used a log-linear formulation (e.g., **c** and **l**):

$$d_{i,m} = e^{a_m \times E_{i,m} + b_m}; a_m \geq 0 \quad (3)$$

2) If $d_{i,m}$ can take both positive and negative values, we used a linear formulation (e.g., $\theta\text{min}$):

$$d_{i,m} = a_m \times E_{i,m} + b_m; a_m \geq 0 \quad (4)$$

This hyperspherical parameterization is mathematically equivalent to a classical linear combination, albeit less intuitive. Compared to the latter, it allowed us to set regularizing priors[65] on the mean and variance of the demographic parameters (through parameters of $a_m$ and $b_m$). This ensured the convergence of the ODE model (Supplementary Information 1.4.3) and controlled parameter trade-offs, but also allowed us to maintain uninformative priors on the parameters $\{\varphi_{m,n}\}$ and avoid making prior assumptions on trait–demography relationships.

### Bayesian Inference of the transfer function parameters

*Likelihood function.* For any given set of parameter values and for each plot temperature $\theta_j$, we initialized the ODE with random positive biomass for all species. The ODE was run for a fixed amount of time steps sufficient to characterize the equilibrium across the prior distribution of the parameters (see Supplementary information 1.4.3). Because of the structure of the pairwise interaction matrix, the ODE model was globally stable for any parameter values: a single simulation was thus sufficient to characterize the equilibrium (see Supplementary information 1.3). We then calculated the likelihood of obtaining the observed set of sampled plant individuals in plot $j$ given the equilibrium state of the ODE model. We assumed that the likelihood follows a multinomial distribution where each species has a probability to be sampled equal to its relative biomass in the community.

*Priors.* We set uninformative priors for all angle parameters $\{\varphi_{m,n}\}$. We further set regularizing priors for all the parameters $a_m$ and $b_m$ which together control the mean and variance of demographic parameters (Supplementary Table 2). Because we calibrated our model using static relative abundances data and we used a likelihood function that follows a multinomial distribution, the demographic parameter $g_i$ and the parameter $b_c$ associated with the demographic parameter vector **c**, were not identifiable (see Supplementary information 1.4). In consequence, they were fixed (for all species $i$, $g_i = 10^{-3.43}$ and $b_c = 10^{-3.8}$, Eqs. 1 and 3). Those parameters would have been identifiable if the dataset included absolute abundance data (parameter $b_c$) or temporal data (demographic parameter $g_i$).

*Posterior estimation.* We used a Differential-Evolution Markov-Chain Monte Carlo algorithm (DEzs MCMC, R-package BayesianTools[66]) to estimate the posterior distribution of the transfer function parameters. We ran eight independent DEzs MCMC chains in parallel for 50000 steps, the posterior was estimated on the last 15000 steps. Convergence of the posterior distribution was assessed with

Gelman–Rubin diagnostics[31] (multivariate psrf was equal to 1.01). Posterior distribution data of the transfer function parameters are available in the Source data.

**Performance assessment**. To evaluate whether the calibrated model performs better than random, we compared its fit to the fits of 200 replicate null models for which the trait data was randomized across species. To generate random functional trait datasets, we shuffled the PCA trait axes values among species and then ran the calibration procedure using these randomized trait data. One chain was run for each randomized dataset. After the burn-in phase (35 000 steps), 95% of the chains displayed a convergence criterion inferior to 1.48. The fits of the calibrated model and of the null models were compared using two metrics: the Deviance information criterion (DIC)[31] and Nagelkerke's pseudo $R^2$ metric[30] which lends itself well to multinomial models and gives an indication of the variance they explain. Nagelkerke's pseudo $R^2$ was calculated from the ratio of a model's posterior likelihood and the likelihood of the hypothesis that all sampled species have equal relative abundance in each plot, as well as the sampling effort. We also computed this pseudo $R^2$ metric at the median of the posterior for each plot individually to access the power of the calibrated model relative to the null model along the studied gradient. Finally, we tested the ability of the calibrated model to predict plant presence/absence in the studied plots by characterizing their ROC curves and AUC scores. To do so, we used the botanical surveys because they are more accurate to infer plant presence/absence compared to transect data[32].

**Comparative analysis**. To illustrate the usefulness of our approach, we compared it to existing correlative approaches. First, we analyzed species abundances along the mean annual temperature gradient using a species distribution model and a set of joint species distribution models[67,68]. The performance of those models was also assessed with the Deviance information criterion, Nagelkerke's pseudo $R^2$, ROC curves, and AUC scores. Second, we analyzed the relationship between mean annual temperature and plant functional traits using the fourth corner analysis[36]. Detailed methods and results are available in the Supplementary information.

All analyses were carried out using the software R 3.5.3, R-packages ade4 1.7.16 and BayesianTools 0.1.7 using the resources of ETH Zürich and the University of Wyoming Advanced Research Computing Center[69].

**Reporting summary**. Further information on research design is available in the Nature Research Reporting Summary linked to this article.

## Data availability

Data are archived in the code repository https://doi.org/10.5281/zenodo.4682287. Source data are provided with this paper.

## Code availability

The code necessary to reproduce the results of this article is archived in the repository https://doi.org/10.5281/zenodo.4682287.

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

## Acknowledgements

LC acknowledges funding from the European Union's Horizon 2020 research and innovation program under the Marie Skłodowska-Curie grant agreement No 840946 (Project "CLIMB"). Data acquisition leading to these results received funding from the European Research Council under the European Community's Seven Framework Program FP7/2007–2013 Grant Agreement no. 281422 (TEEMBIO) granted to W. Thuiller (CNRS-Univ. Grenoble Alpes). We thank T. Ott (University of Regensburg) and B. Flück (ETH Zürich) for coding assistance. LP acknowledges funding from the Swiss National Science Foundation (SNSF) project no. 162604.

## Author contributions

LC, FH and LP conceived and designed the study. LC implemented the models and conducted the main data analysis. LC and HL studied the mathematical properties of the model. LC and MP conducted the comparative analysis. LC, FH, DL, HL, MP, DS and LP discussed all aspects of the research and contributed to writing and revising the paper.

## Competing interests

The authors declare no competing interests.
