## [Peer Review File · Nature Communications]

Reviewer comments, first round:

Reviewer #1 (Remarks to the Author):

The authors test a long standing assumption about the link between traits and demography that would be crucial to understanding plant population and community dynamics. Importantly, these dynamics could be inferred from static data, which overcomes a massive hurdle of data collection for community demography. Hence the framework they propose for linking traits to demography is surely of paramount interest and importance for understanding plant communities and predicting their future. Also, because the method is fairly straightforward, with some nice intuitive features, I expect that it will be readily adopted by the research community. The description of the transfer function needs some improvement, but otherwise this is a very strong paper.

72: I think a few more lines should be used to explain the model in more detail. E.g, its not clear how one would choose the right combinations of sin and cos of additional angles if more traits were included. (It doesn't need to be lengthy, but since few ecologists will know this spherical parameterization, they shouldn't have to go look elsewhere to find out what it is.) Relatedly, how about writing out the full model for $d_{i,j}$ relating these angles and the mean and sd?

95: I think people may be confused by referring to something like min temp tolerated as a demographic trait. Most would think of demography as individual survival, growth, and reproduction I think. I realize that you could equally well model those demographic rates the same way, but there may be a better term here. I've also heard these quantities you refer to as demographic rates also called 'traits' elsewhere.

168: Surely there must be some art in choosing how/which demographic quantities to model to avoid tradeoffs in parameter estimation, right? While the short format of the article may mean that this can't go in the main text, it probably deserves significant discussion.

274: This section on the transfer function isn't very clear. I don't see E_{ij} defined. It appears that equations 1 and 2 are equalities, not distributions as they are referred to in the text. Its not clear why d_{ij} should have a site dependence for a quantity like minimum temperature tolerated. Isn't this an attribute of a species, and therefore wouldn't depend on site?

306: So does the ability to fit this model depend on the assumption that the community Abundance distribution is at equilibrium?

Please include sample code for fitting the model.

Fig 4: While there are some general trends kind of evident here, its not clear what the takeaway should be from the figures. Sure, there are more grasses at higher elevation, but that doesn't show off any particular success of the model. It doesn't seem as though the accuracy or precision of the model's ability to predict community abundance is provided anywhere. That seems a better topic for this figure.

Reviewer #2 (Remarks to the Author):

Nat. Comm. 236557_0_merged_1578326320

General

This paper uses an inverse modelling approach to derive species-specific demographic parameters from species-specific functional trait data using species abundances data and data on the environment, here a single environmental variable. The key novelty lies in the use of traits so as to

allow usage of existing inverse modelling approaches to species-rich communities. I recommend the paper.

I have many detail comments that needs to be addressed. The most urgent ones are on the use of spherical parametrization, on how the trait data are randomized and on a comparison of the fit with a SDM-fit. Note that a relatively bad fit should in my view not necessarily hamper publication of the paper; the approach has worth in its own. The "fit well" in the abstract is presumably overoptimistic.

Details

L9 I suggest: of models-> of a process-based model

L9 "by calibrating a transfer function". Add "by inverse modelling".

L10 "a high-diversity mountain grassland community model" It is not the model that is high-diversity. Why not change to "a modified Lotka-Volterra model of a high-diversity mountain grassland community". I remark that "community model" is (re)defined in the paper to be process-based, but that cannot be clear in the abstract.

L11 "predicted"? The calibrated transfer function IS the fitted T-D relationship. Simplify. Perhaps remove the term transfer function at all; it can mean anything, including inverse model.

L57 The change with environmental context is beyond the scope of the model presented, I presume. So the next "We solve this" is in my view an overstatement.

L61-62 I would suggest [in steno]: we define D as a parametric function of the empirical functional traits T. [Why otherwise use the word "define"? In math, one defines D as demographic parameters..., and not the other way round...]

L63-64. "We use a spherical parametrization of the correlation [25]," The word "correlation" suggest something symmetric, whereas as Eq[1] on Line 70 shows that d is a function of three [synthetic] traits. I have nothing particularly against this special parametrization a priori. But please motivate it, perhaps in a Supplement. The spherical parametrization is meant for parameters in a matrix that should be positive definite, according to [25]. The Cholesky factor then is parametrized in this way. Is this useful/appropriate for D, being parameters of the Lotka-Volterra model? Is it needed for the model to have a single stable equilibrium? The later choice that c is the intraspecific competition rate minus I would also interfere with such a motivation.

L64 "motivated by ecology theory [26]" Can you indicate which aspect in [26] motivated your choice? I did not see it after reading [26]. Wrong reference?

L73 I find it strange to say "sd and mean". One normally says "mean and sd"

L74-75. Any linear regression coefficient in a regression with standardized predictors and response is interpretable as a partial correlation/regression coefficient. I did not find the Supplement on this convincing. The parabolic curve in the right-most figure would even bother me. Two parameters for three predictors worries me as well.

L88 [25,26] wrong reference!

L92 Eq 2 is far below here.. Note that there are even two Eq2!

L70 I would consider the details of the trait-demographic rates relationship as, indeed, details, compared to the Lotka-Volterra model and its calibration. So why is equation (1) so important that this one is in the introduction. Said otherwise, is the spherical parametrization the key novelty? I think the key novelty is the trait-based link from dynamic/process-based models to static data. Reconsider.

The order of the PCA used in Eq1 is now going to matter, is it not? If one interchanges PCA1 and PCA2, the resulting fits might change?

L95 I would suggest "within-species" -> species-specific

L102 "random trait data" -> randomized trait data?

L102 and L107 You use in the R code

```
> # Randomization of trait values
```

```
> tr <- apply(tr, 2, sample)
```

This randomizes each synthetic trait individually. This destroy their covariance matrix (no longer I). It would be much more logical to randomize synchronously, by `tr <- tr[sample(nrow(tr)),]`.

L105-106 Improved compared to what. Interchange with 106-107?

L106-109. Note that that DIC (or any information criterion) is not a goodness of fit statistic (see e.g. example in Lesaffre & Lawson 2012 Bayesian Biostatistics p269). The fact that DIC of the data as observed is lowest, does not say that the model fits well.

L112 Whereas Supplementary Fig.5 is about comparison of modeled vs observed data, the Supplement says nothing (?) about competitive exclusion. Clarify.

L226 "Trait measurements were averaged by species" Fine, that would perhaps allow use of the method with traits from trait data bases. But, would it have been an advantage to use plot-specific trait measurements per species?

L247 "to simplify" -> "and was simplified by setting equal the inter-specific competition rates" or something along this line.

L248 With these simplifications, can it be shown that there is a unique stable equilibrium for each set of parameter values? [so that the single random start in the inverse modelling approach is warranted].

L252. Add: The rhs of Eq2 is the intrinsic relative biomass growth rate - incorporating reproduction, mortality and individual biomass growth - of a species i in a site j .

L260 "relative biomass growth rate": use the same wording in L 253 and L 260 and L265. Note that the font of the parameters differs between the equations and the text. Adapt.

L256. And: below temp-min, the growth is negative. Is that what you want?

L257 within-species-> species-specific.

L271 "Depending on the strength of trade-off among traits". But the model in this section has no traits!

L280. L280 is still main text, isn't it?

L283-285. And is this important/relevant for the model?

L289. No "e.g." here: you can be specific as there are few possibilities.

L290 Eq 1. Change equation number. E_{ij} is undefined and d_{ij} appears to be a different thing (location/scale) than d_{ij} in L70. The proportionality sign in L70 accounts for a_i , but not b_i .

L292 renumber equation.

L299-300. The remark on proportionality applies to the parameters g, c, l in Eq(2) on line 250. Doesn't that mean that the parameter a_i in line 290 is redundant and can be set to any positive value? The Eq(2) on line 250 contains a parameter c . What are then a_c and b_c here; I would not term these mean and variance of the rate c . Rethink terminology.

L303-307 Rephrase taking the following comments in mind:

L303 likelihood -> likelihood of the observed abundances

L304 Add something like: we need the equilibrium biomasses of species, which was by calculated by starting the ODE with random positive biomass.... etc. Note that you also mix biomass with number of individuals. Explain more clearly. Perhaps change the term biomass to number of individuals?

L305. For a statistician it does not help that biomass in the ODE is P (which suggests something relative or adding to 1). For the multinomial you need to divide by the plot-specific total (as done OK in the R-code).

L305 probability: the equilibrium is in terms of biomass, which is then divided by the total biomass to give fractions. These fractions are taken as the probabilities of a multinomial distributions; the data are the observed numbers. Note that this is a simple likelihood model; it does not account for overdispersion. For example, when negative binomial counts are analyzed in the relative version, the result is not a multinomial.

L307 In the multinomial distribution the data "y" is not relative. The probabilities are.

L317-318. Rephrase. You estimate the posterior distribution by drawing parameter combinations for the posterior sample.

L319-320. Say how you go from the PCA scores to the trait values [by the biplot or innerproduct rule; Gabriel 1971, I guess]. Or, do you mean that I should read "analyzed" as "calculated correlation coefficients between observed T and D" ?

L325 See above on randomization.

Figure 2 legend. I find terming Panel (A) a correlogram weird; it normally refers to figures with lags on the x-axis. Say more simple. Does Panel (A) contain correlations? See comment at L 319-320.

To improve clarity-> For clarity.

Figure 4. In my opinion, there is no reason to use a scaled temperature instead of the real scale.

Additional remarks/questions

It would be nice to have a figure comparing the fit of each species abundance against temperature (SDMs) with the fits by the model in Fig. 4

There are two plots per temperature. There are only 8 different temperatures. How do or could these numbers influence the inverse modelling approach? The approach seems to be primarily a

species-level approach. The number of sites/replications are rather implicit.

Can you say which of the three synthetic traits or of the measured traits is most important?

Supplement 236557_0_supp_4309524_q2q4t6.pdf:

"can be assimilated" say more simple (twice).

On p3. The formula for y is in error I suppose. Φ has indices $i, 1$ or just 1 or 2 ?

"the two parameters can be assimilated to a transformed value of the correlation or partial"

Assimilated? Understood as? Is the parabola in Supp Fig 3 not weird/strange. How to interpret that one?

After finishing the review (and the 10-day deadline for the review) I got a new insight the next day that (1) indeed, the model requires a specialized (constrained) parametrization (2) that led also to the conclusion that the spherical parametrization used in the paper does not solve the problem at all; the constraints are at the wrong side of the equation. (3) that there exist a nice solution for the parametrization issue.

The authors can contact me if they would like to discuss these points.

Wageningen, 28 January 2020

Cajo ter Braak cajo.terbraak@wur.nl

Reviewer #3 (Remarks to the Author):

There is a great challenge in reducing complex ecological systems to simpler approximations so that models can better capture and predict system dynamics.

This paper is clearly established in this program, and the transfer function seems to be an approach with the potential to advance that. The manuscript as written, however, seems to straddle several fields (community ecology, statistical methods, process modeling) without comprehensively advancing them as would be appropriate for Nature Communications.

I worry that this manuscript's potential impact relies heavily on 1) traits as being indicators of demography (even though this is addressed, the manuscript does not correct it explicitly), 2) community models as being able to capture and predict system changes through the inverse estimates of parameters, and 3) a grassland analysis with moderately encouraging results as demonstrating the first two points.

This falls short of a comprehensive demonstration of a method, and fails to reveal a story in community ecology that establishes the study as a groundbreaking work. I do think the approach is worthwhile and it would be interesting to see it applied with other datasets in other contexts.

There are a number of typos and grammatical mistakes that should be corrected.

Dear reviewers,

We would like to thank the three reviewers for the constructive comments on our manuscript. We first detailed the main methodological changes, as suggested by Reviewer #1 and #2. We then answer point by point the other (non-methodological) comments raised by the reviewers, following each original comments.

Methodological changes

1) After exchanging with Reviewer #2, we now better account for the colinearities inherent to the Lotka-Volterra model. In consequence, we fixed one demographic rate (demographic rate “g”, growth rate sensitivity to temperature). The reasoning behind this is that species abundances at equilibrium do not vary when our four initial demographic rates vary in proportion to each other. We further fixed the parameter that controlled the magnitude of the demographic rate “c” (b_c) because relative abundances of species also stay constant if species abundances vary in proportion to each other. Thus if left unfixed, this parameter posterior distribution was fully correlated to the parameter (b_l) posterior distribution. In the end, our new version of the model now estimates a total of 11 parameters that described the variation of three demographic rates across species.

2) Reviewer #1 and #2 expressed concerns about our assessment of model performance, as DIC was deemed insufficient. We faced the initial difficulty that our likelihood follows a multinomial law incompatible with classical R^2 metrics. As a way around it, we now use the pseudo- R^2 metric as defined by Nagelkerke (1991) alongside the DIC metric. This pseudo- R^2 metric is based on the ratio of our predicted likelihood on the likelihood of a null hypothesis and the sample size. Its upper limit is 1 when there is a perfect fit to the data. Here we chose the null hypothesis: “all species have equal relative abundance in every plot” as a baseline. Pseudo- R^2 were calculated for each plot individually and for the whole dataset to give a complete image of our model performance. We further calculated those metrics for the “randomized functional trait” models. This new analysis of the results is displayed in the updated Figure 4.

3) In supplementary materials, we examined if the change in PCA axes ordering can affect parameter estimation, we conclude that it does not. We also examine if including a fourth trait axis would change our results, we conclude that it does not.

4) Following the suggestion of Reviewer #2, we compared our approach to a “stacked SDM”-like approach (see complementary study #1 for detailed results). We calculated the correlation coefficient between the predicted species relative abundance from both approaches. Our main conclusion is that both our model and a stacked SDM overall suggest similar trends of species abundance along the temperature gradient: both predicted relative abundances are positively correlated across all species and plots as well as within individual species. However the quality of that correlation was much better when focusing only on well sampled (aka dominant) species. This is partly due to the fact that our model tends to produce false positives

but also because the SDM approach produces false negatives (a likely consequence of modeling species individually). This can be linked to the limitations of the sampling scheme : as we already highlighted in Supplementary material 2.3, an absence of sampled individuals of a species in a plot can either show its lack of local dominance rather than its strict absence.

We leave the decision to include that study in the supplementary materials to the editor and reviewers. We would argue against : while informative, this small comparison is rough and is far from being an adequate test of the merits of stacked SDM approaches vs. our new approach. There are strong limitations to the use of SDMs to model each species individually in our dataset: few plots and few sampled individuals for most species were only sampled a few times. This likely means that our SDM-like analysis suffers from overfitting. By including it, we feel that it would suggest that the SDM approach gives the ‘best’ answer that our modeling approach should tend to; this would be, in our opinion, misleading.

5) We took the opportunity of the revision to check some assumptions we made in the first version of the analysis, we thus included two more plots in the analysis (at high elevation) that were previously removed. We further modify our method to produce positive demographic rates (Equation 2): we now use an exponential transformation instead of an ‘expit’ transformation because it considerably improved the fit of our model. We further refined the analysis priors to improve parameter estimations and speed model convergence.

REVIEWERS’ COMMENTS

Reviewer #1 (Remarks to the Author):

The authors test a long standing assumption about the link between traits and demography that would be crucial to understanding plant population and community dynamics. Importantly, these dynamics could be inferred from static data, which overcomes a massive hurdle of data collection for community demography. Hence the framework they propose for linking traits to demography is surely of paramount interest and importance for understanding plant communities and predicting their future. Also, because the method is fairly straightforward, with some nice intuitive features, I expect that it will be readily adopted by the research community. The description of the transfer function needs some improvement, but otherwise this is a very strong paper.

Thank you.

72: I think a few more lines should be used to explain the model in more detail. E.g, its not clear how one would should choose the right combinations of sin and cos of additional angles if more traits were included. (It doesn't need to be lengthy, but since few ecologists will know this spherical parameterization, they shouldn't have to go look elsewhere to find out what it is.) Relatedly, how about writing out the full model for $d_{\{i,j\}}$ relating these angles and the mean and sd?

We know wrote the full equation for any number of traits. We kept the full equation separated in two because, technically, any monotonous transformation of $E_{i,m}$ could be envisioned depending on what is suspected to be an adequate distribution of the demographic parameter across species. For instance, between the first version and the present version, we changed the transformation function for positive demographic parameters from an expit to an exponential distribution because it was given more support from the data. We know state this.

l. 294: ‘However, any monotonically increasing transformation of $E_{i,m}$ could be employed depending on data support.’

274: This section on the transfer function isn't very clear. I don't see E_{ij} defined. It appears that equations 1 and 2 are equalities, not distributions as they are referred to in the text. It's not clear why d_{ij} should have a site dependence for a quantity like minimum temperature tolerated. Isn't this an attribute of a species, and therefore wouldn't depend on site?

We agree that it would be clearer. Some demographic rates (l and c in the article) need to be positive whereas T_{min} can take both negative and positive values. We thus need two different ways of transforming the results of Equation 2. This led us to the current layout, we improved from the previous version by calling the result of Equation 2 ' E_{ij} ' and qualify it explicitly as an "intermediate expression" that is then either transformed using Equation 3 or Equation 4. Note that this whole section was moved to the methods, which is better to make the connexion between Equation 2, 3 and 4.

Demographic parameters do not vary across sites. d_{ij} in that paragraph referred to the demographic parameters i of species j . We guess that it was confused with the subscripts in Equation 1 where B_{ij} is the biomass of species i in site j . We changed the subscript denomination. ' i ' now refers consistently to species, ' j ' to sites, ' m ' to demographic parameters' and ' n ' to trait throughout the paper.

95: I think people may be confused by referring to something like min temp tolerated as a demographic trait. Most would think of demography as individual survival, growth, and reproduction I think. I realize that you could equally well model those demographic rates the same way, but there may be a better term here. I've also heard these quantities you refer to as demographic rates also called 'traits' elsewhere.

We now call them "demographic parameter" consistently throughout the manuscript. We kept the expression "demographic rates" only in some introductory sentences to refer to other studies. It is further explicitly defined in the introduction:

1.44: Our idea is to establish a **transfer function** that links empirically measured functional traits with the parameters of community models that describe species demography, hereafter called "demographic parameters".

168: Surely there must be some art in choosing how/which demographic quantities to model to avoid tradeoffs in parameter estimation, right? While the short format of the article may mean that this can't go in the main text, it probably deserves significant discussion.

This is indeed our argument in that paragraph, it was however diluted in the successive text modifications. We reformulated that paragraph to better express our ideas.

1.158: "The only requirement is that the process must be formulated as a function of attributes that can be linked to species-specific functional traits. However, some processes, typically biotic interactions, are modeled as attributes of two or more species¹⁹ and would need to be reformulated. Several theoretical studies have shown a path forward, for instance by expressing pairwise interaction coefficients from Lotka-Volterra competition models as functions of species ability to acquire resources^{44,45}. Empirical studies have further suggested how pairwise interactions among species can be modeled from their functional traits^{18,46} and could inspire new theoretical frameworks for modeling biotic interactions.

Beyond the scope of our community model, one could also consider other features of biotic interactions such as niche partitioning among species⁴⁷, positive biotic interactions⁴⁸ (e.g. facilitation) or trophic interactions⁴⁶."

306: So does the ability to fit this model depend on the assumption that the community abundance distribution is at equilibrium?

It does not. One can imagine the principle of the transfer function applied to datasets that include the temporal dynamic of communities. However as our data does not provide temporal trends, so we felt that the most reasonable assumption for our study was to assume that the studied communities are at equilibrium in regard of the current environmental conditions. We know spell out this assumption:

1.313: For the sake of simplicity and in the absence of temporal data, we assumed that the observed communities were at equilibrium with the current climatic conditions.

Please include sample code for fitting the model.

It is included.

Fig 4: While there are some general trends kind of evident here, it's not clear what the takeaway should be from the figures. Sure, there are more grasses at higher elevation, but that doesn't show off any particular success of the model. It doesn't seem as though the accuracy or precision of the model's ability to predict

community abundance is provided anywhere. That seems a better topic for this figure.

Figure 4 has been completely modified to better show this. Using pseudo R², we know details for each observed community how good is the model fit.

Reviewer #2 (Remarks to the Author):

Nat. Comm. 236557_0_merged_1578326320

General

This paper uses an inverse modelling approach to derive species-specific demographic parameters from species-specific functional trait data using species abundances data and data on the environment, here a single environmental variable. The key novelty lies in the use of traits so as to allow usage of existing inverse modelling approaches to species-rich communities. I recommend the paper.

I have many detail comments that needs to be addressed. The most urgent ones are on the use of spherical parametrization, on how the trait data are randomized and on a comparison of the fit with a SDM-fit. Note that a relatively bad fit should in my view not necessarily hamper publication of the paper; the approach has worth in its own. The “fit well” in the abstract is presumably overoptimistic.

Thanks. We joined a short study comparing pattern with a SDM fit. Please see above.

Details

L9 I suggest: of models-> of a process-based model

L9 “by calibrating a transfer function”. Add “by inverse modelling”.

Corrected

L10 “a high-diversity mountain grassland community model” It is not the model that is high-diversity. Why not change to “a modified Lotka-Volterra model of a high-diversity mountain grassland community”. I remark that “community model” is (re)defined in the paper to be process-based, but that cannot be clear in the abstract.

Corrected

L11 “predicted”? The calibrated transfer function IS the fitted T-D relationship. Simplify. Perhaps remove the term transfer function at all; it can mean anything, including inverse model.

We changed that sentence to : “The calibrated trait-demography relationships were amenable to ecological interpretation, and led to species abundances which fit well to the observed community structure.”

L57 The change with environmental context is beyond the scope of the model presented, I presume. So the next “We solve this” is in my view an overstatement.

Removed.

L61-62 I would suggest [in steno]: we define D as a parametric function of the empirical functional traits T . [Why otherwise use the word “define”? In math, one defines D as demographic parameters..., and not the other way round...]

Changed to (and moved to the methods, l. 272) : “Given a matrix of N empirical functional traits $\mathbf{T} = \{t_{i,n}\}$, where $t_{i,n}$ is the known value of functional trait n of species i , and a matrix of unknown demographic parameters $\mathbf{D} = \{d_{i,m}\}$, where $d_{i,m}$ is the value of demographic parameter m of species i , the transfer function specifies the mathematical link between the two. “

L63-64. “We use a spherical parametrization of the correlation [25],” The word “correlation” suggest something symmetric, whereas as Eq[1] on Line 70 shows that d is a function of three [synthetic] traits. I have nothing particularly against this special parametrization a priori. But please motivate it, perhaps in a Supplement. The spherical parametrization is meant for parameters in a matrix that should be positive definite, according to [25]. The Cholesky factor then is parametrized in this way. Is this useful/appropriate for D , being parameters of the Lotka-Volterra model? Is it needed for the model to have a single stable equilibrium? The later choice that c is the intraspecific competition rate minus 1 would also interfere with such a motivation.

We used those references too casually. [25] actually focuses on the parametrization of correlation matrices among variables, whereas in our case it is between two sets of variables. Because of that the Cholesky factorization and positive-definite matrices are not relevant here, the matrix form with the trigonometric functions does not even have to be square if the number of trait and the number of demographic parameters is different.

We rather used this reference because like our work, it is rooted in a geometric interpretation of correlations coefficients that we now explain better. We further note cite this work that is also dealing with correlation matrices like [25] but focuses more on the “correlation as angles” idea, we believe it would be a better illustration:

Rapisarda, F., Brigo, D. & Mercurio, F. Parameterizing correlations: a geometric interpretation. *IMA Journal of Management Mathematics* **18**, 55–73 (2007).

We improved this part of the article with the following paragraphs:

l. 56 (introduction): “A first set defines a correlation structure between the unknown demographic parameters and the empirical functional traits^{23,24}. (see methods for more details). These transfer parameters can be individually interpreted as (partial) correlation coefficients between each functional trait and each demographic parameter (see Supplementary materials). The second set of parameters defines the mean and standard deviation of each demographic parameter across species.”

l. 281 (methods): “We used a geometric parameterization of the correlations²⁴ between demographic parameters and functional traits: the coefficients of the linear expression linking the species demographic parameters values $d_{i,m}$ to the functional trait values $t_{i,n}$ can be viewed as coordinates lying on a unit hypersphere of dimension N . The coefficients can be then defined by $N-1$ angles ($\varphi_{m,n}$). The matrix of angles captures the correlation structure between functional traits and demographic parameters across species (Supplementary materials).”

L64 “motivated by ecology theory [26]” Can you indicate which aspect in [26] motivated your choice? I did not see it after reading [26]. Wrong reference?

[26] was an ecology paper that made use of this parametrization, we aimed at providing it as “in-house” example. But you are right, it is not a good reference to illustrate our argument, as the approach is only mentioned in passing in the methods of the paper. We dropped it.

L73 I find it strange to say “sd and mean”. One normally says “mean and sd”

Corrected.

l.59: The second set of parameters defines the mean and standard deviation of each demographic parameter across species.

l.297: Both expressions depend on additional parameters a_m and b_m that together control the mean and standard deviation of $d_{i,m}$ across species.

L74-75. Any linear regression coefficient in a regression with standardized predictors and response is interpretable as a partial correlation/regression coefficient. I did not find the Supplement on this convincing. The parabolic curve in the right-most figure would even bother me. Two parameters for three predictors worries me as well.

We made a mistake with that graph by forgetting half of the distribution of ϕ_2 (it was distributed across $[0, \pi]$ instead of $[0, 2\pi]$). This was corrected.

Once the correlation coefficient with T1 is set with ϕ_1 and the partial correlation to T2 is fixed with ϕ_2 , then the remaining variance is necessarily fully explained by the third axis that is a necessary constrain to keep the variance of the expression E_{ij} constant across species.

Note also that our if we had specify a classical linear combination:

$d = x_1 * t_1 + x_2 * t_2 + x_3 * t_3 + x_4$, we would have had four parameters to estimate. Exactly like in our approach that uses the parameters ϕ_1 , ϕ_2 , a and b . So we are not removing a degree of freedom with our approach. Another way to compare a classical combination with our approach is to think about it as if we estimate the x_1 , x_2 and x_3 parameter while keeping the total $x_1^2 + x_2^2 + x_3^2$ value as a single parameter to be evaluated or kept constant if needed (like we did for the demographic parameter c).

L88 [25,26] wrong reference!

Corrected

L92 Eq 2 is far below here.. Note that there are even two Eq2!

We moved all the mathematical bits to the methods and corrected the equation numbering.

L70 I would consider the details of the trait-demographic rates relationship as, indeed, details, compared to the Lotka-Volterra model and its calibration. So why is equation (1) so important that this one is in the introduction. Said otherwise, is the spherical parametrization the key novelty? I think the key novelty is the trait-based link from dynamic/process-based models to static data.

We agree. We now put the details of the transfer function in the methods, we instead make a brief summary of the concepts behind the transfer function in the introduction and discuss further the rationale of using “correlation parameters” and “mean/average parameters” instead of a classical linear combination (I.XX).

The order of the PCA used in Eq1 is now going to matter, is it not? If one interchanges PCA1 and PCA2, the resulting fits might change?

We now check this, the results are available in the supplementary materials.

L95 I would suggest “within-species” -> species-specific

Based on our email, this demographic parameter was removed from the analysis and kept constant across species.

L102 “random trait data” -> randomized trait data?

Corrected.

L102 and L107 You use in the R code

```
> # Randomization of trait values
```

```
> tr <- apply(tr, 2, sample)
```

This randomizes each synthetic trait individually. This destroys their covariance matrix (no longer I). It would be much more logical to randomize synchronously, by `tr <- tr[sample(nrow(tr)),]`.

This was corrected in the new revised analysis.

L105-106 Improved compared to what. Interchange with 106-107?

L106-109. Note that that DIC (or any information criterion) is not a goodness of fit statistic (see e.g. example in Lesaffre & Lawson 2012 Bayesian Biostatistics p269). The fact that DIC of the data as observed is lowest, does not say that the model fits well.

We added Nagelkerke’s pseudo-R² metric.

L112 Whereas Supplementary Fig.5 is about comparison of modeled vs observed data, the Supplement says nothing (?) about competitive exclusion. Clarify.

Corrected to the more explicit and factual “it did not predict well species absences“

L226 “Trait measurements were averaged by species” Fine, that would perhaps allow use of the method with traits from trait data bases. But, would it have been an advantage to use plot-specific trait measurements per species?

Probably. But the traits were only measured on the individuals sampled along the transects. So the sampling was not adequate to characterize species trait in each plots (for instance if a given species was not sampled in a plot or too few individuals were).

L247 “to simplify” -> “and was simplified by setting equal the inter-specific competition rates” or something along this line.

This is explicitly stated later when each demographic parameter is detailed. We feel that this wording would suggest all the off-diagonal elements of the pairwise competition rates are equal.

L248 With these simplifications, can it be shown that there is a unique stable equilibrium for each set of parameter values? [so that the single random start in the inverse modelling approach is warranted].

Yes. The demonstration is now included in the supplementary materials.

L252. Add: The rhs of Eq2 is the intrinsic relative biomass growth rate - incorporating reproduction, mortality and individual biomass growth - of a species *i* in a site *j*.

We added part of Equation 2 in the text.

L260 “relative biomass growth rate”: use the same wording in L 253 and L 260 and L265. Note that the font of the parameters differs between the equations and the text. Adapt.

L256. And: below temp-min, the growth is negative. Is that what you want?

Yes. This is now made explicit : “below that threshold species j goes extinct.”

L257 within-species-> species-specific.

This was removed as g is now kept constant across species.

L271 “Depending on the strength of trade-off among traits”. But the model in this section has no traits!

Corrected to “demographic parameters”.

L280. L280 is still main text, isn't it?

This was removed as we restructured the explanation of the transfer function.

L283-285. And is this important/relevant for the model?

Maybe. It facilitates the approach as we know that a single ODE simulation is sufficient to characterize the equilibrium of the ODE. We leave the question open to whether this is a necessary features of the model.

This is now touched upon in the discussion :

L168: “In such more complex models, several equilibria could exist, which might better reflect local heterogeneity of plant communities and the transient nature of many ecosystems. This would likely require a larger community dataset with more community sampled in similar abiotic environments and would also raise new analytical challenges as simulation approaches such as ODEs might not be appropriate to efficiently characterize those multiple equilibrium states⁴⁹.”

L289. No “e.g.” here: you can be specific as there are few possibilities.

Corrected

L290 Eq 1. Change equation number. E_{ij} is undefined and d_{ij} appears to be a different thing (location/scale) than d_{ij} in L70. The proportionality sign in L70 accounts for a_i , but not b_i .

L292 renumber equation.

Corrected.

L299-300. The remark on proportionality applies to the parameters g, c, l in Eq(2) on line 250. Doesn't that mean that the parameter a_i in line 290 is redundant and can be set to any positive value? The Eq(2) on line 250 contains a parameter c . What are then a_c and b_c here; I would not term these mean and variance of the rate c . Rethink terminology.

We change the sentence to “which together control the mean and variance of the demographic parameters” to remove the ambiguity.

L303-307 Rephrase taking the following comments in mind:

L303 likelihood -> likelihood of the observed abundances

Corrected to : “To calculate the likelihood of the observed relative abundances”

L304 Add something like: we need the equilibrium biomasses of species, which was by calculated by starting the ODE with random positive biomass.... etc. Note that you also mix biomass with number of individuals.

Explain more clearly. Perhaps change the term biomass to number of individuals?

L307 In the multinomial distribution the data “ y ” is not relative. The probabilities are.

The values given by the model are not integers, so using “individual” might sound misleading. We are not sure what the reviewer found problematic in the previous version of the paragraph with the remark that “the data “ y ” is not relative”.

We changed this paragraph as follows and hope it clears any ambiguities:

L. 312: “For any given set of model parameters and for each empirical temperature θ_j , we initialized the ODE with random positive biomass for all species. For the sake of simplicity and in the absence of temporal data, we assumed that the observed communities were at equilibrium with the current climatic conditions. Thus, the ODE model was run until equilibrium to calculate the likelihood of obtaining the observed set of sampled plant individuals along the transects. We assumed that this likelihood follows a multinomial distribution where each species has a probability to be sampled equal to its relative biomass in the community.”

L305. For a statistician it does not help that biomass in the ODE is P (which suggests something relative or adding to 1). For the multinomial you need to divide by the plot-specific total (as done OK in the R-code).

Changed to B for “biomass”

L305 probability: the equilibrium is in terms of biomass, which is then divided by the total biomass to give fractions. These fractions are taken as the probabilities of a multinomial distributions; the data are the observed numbers. Note that this is a simple likelihood model; it does not account for overdispersion. For example, when negative binomial counts are analyzed in the relative version, the result is not a multinomial. As a side note, we have tested the more complex likelihood function of Damgaard 2015 (10.1016/j.ecoinf.2015.10.006) that compounds a multinomial law and a Dirichlet distribution with an overdispersion parameter. That parameter was calibrated to values that indicates only a small intra-plot correlation (δ 95% CI [0.049-0.06]), calibrated trait-demographic relationships were only changed marginally and the comparison of predicted relative abundances showed a very strong correlation ($r = 0.93$). The only notable difference (which did not affect significantly the output of the model) was with the small relative abundances. Where the multinomial model predicted absences or very small relative abundances for some species in some sites, the more complex model predicted higher (small) relative abundances. We previously identified that the multinomial model tend to produce false positives, it seems that the more complex model tend to worsen that behavior. (see figure S5.jpeg)

L317-318. Rephrase. You estimate the posterior distribution by drawing parameter combinations for the posterior sample.

Changed to : “To study the output of our calibrated community model, we drew parameter combinations from the posterior sample to estimate the posterior distribution of species demographics rates”

L319-320. Say how you go from the PCA scores to the trait values [by the biplot or innerproduct rule; Gabriel 1971, I guess]. Or, do you mean that I should read “analyzed” as “calculated correlation coefficients between observed T and D” ?

No. We calculated the correlation with the raw, original empirical values of the functional traits that were used to generate the PCA axes. The sentence now reads as:

l.325: “We analyzed the posterior demographic parameters in relationship to the observed species functional traits values used to generate the PCA trait axes, rather than to the PCA trait axes themselves. We assessed the posterior correlation among the calibrated demographic parameters, the correlation between demographic parameters and empirical functional traits and the distribution of demographic parameters among functional groups (forbs, grasses, legumes and shrubs).”

L325 See above on randomization.

Fixed.

Figure 2 legend. I find terming Panel (A) a correlogram weird; it normally refers to figures with lags on the x-axis. Say more simple. Does Panel (A) contain correlations? See comment at L 319-320.

To improve clarity-> For clarity.

Changed to : “Calibrated correlation between demographic parameters and empirical functional traits (A) and among demographic parameters (B) as calibrated by our procedure. The graphic display the correlation coefficients (...)”

Figure 4. In my opinion, there is no reason to use a scaled temperature instead of the real scale.

Additional remarks/questions

We now used the non scaled temperature.

It would be nice to have a figure comparing the fit of each species abundance against temperature (SDMs) with the fits by the model in Fig. 4

See above our response.

There are two plots per temperature. There are only 8 different temperatures. How do or should these numbers influence the inverse modelling approach? The approach seems to be primarily a species-level approach. The number of sites/replications are rather implicit.

The number of sites and replication is stated in the methods. The model does not take into account intra-site heterogeneity, so as this stage, a intra-site effect would not be modeled by the approach. This could be envision for future developments.

We discuss it quickly in the discussion: (l. 167) “In such more complex models, several equilibria could exist, which might better reflect local heterogeneity of plant communities and the transient nature of many ecosystems. This would likely require a larger community dataset with more community sampled in similar

abiotic environments and would also raise new analytical challenges as simulation approaches such as ODEs might not be appropriate to efficiently characterize those multiple equilibrium states⁴⁹.”
We would disagree that the approach is species-level, as both the sampling scheme (that was species-blind) and the choice of a multinomial likelihood function can be typically characterized as community-level. Especially in comparison to SDM approaches.

Can you say which of the three synthetic traits or of the measured traits is most important?

Supplement 236557_0_supp_4309524_q2q4t6.pdf:

“can be assimilated” say more simple (twice).

On p3. The formula for y is in error I suppose. Φ has indices $i, 1$ or just 1 or 2 ?

“the two parameters can be assimilated to a transformed value of the correlation or partial” Assimilated?

Understood as? Is the parabola in Supp Fig 3 not weird/strange. How to interpret that one?

We made a mistake with the third panel (see above). The Φ parameters are now named Φ_1 and Φ_2 in the legend (as in the formula). The legend now reads as:

“The figure shows that the two parameters can be understood as transformed values of the correlation or partial correlation coefficients between y and t_1 , t_2 and t_3 .”

After finishing the review (and the 10-day deadline for the review) I got a new insight the next day that (1) indeed, the model requires a specialized (constrained) parametrization (2) that led also to the conclusion that the spherical parametrization used in the paper does not solve the problem at all; the constraints are at the wrong side of the equation. (3) that there exist a nice solution for the parametrization issue.

The authors can contact me if they would like to discuss these points.

Thanks for letting us contact you directly. It was helpful and instructive and we hope that you appreciate the new version of the analysis.

Wageningen, 28 January 2020

Cajo ter Braak cajo.terbraak@wur.nl

Reviewer #3 (Remarks to the Author):

There is a great challenge in reducing complex ecological systems to simpler approximations so that models can better capture and predict system dynamics.

This paper is clearly established in this program, and the transfer function seems to be an approach with the potential to advance that. The manuscript as written, however, seems to straddle several fields (community ecology, statistical methods, process modeling) without comprehensively advancing them as would be appropriate for Nature Communications.

I worry that this manuscript’s potential impact relies heavily on 1) traits as being indicators of demography (even though this is addressed, the manuscript does not correct it explicitly), 2) community models as being able to capture and predict system changes through the inverse estimates of parameters, and 3) a grassland analysis with moderately encouraging results as demonstrating the first two points.

This falls short of a comprehensive demonstration of a method, and fails to reveal a story in community ecology that establishes the study as a groundbreaking work. I do think the approach is worthwhile and it would be interesting to see it applied with other datasets in other contexts.

There are a number of typos and grammatical mistakes that should be corrected.

We made significant improvements for the second version of this manuscript. We hope that the new version convinces you of the value of this study for the field.

Reviewer comments, second round:

Reviewer #1 (Remarks to the Author):

Overall, I think this is a great set of ideas, but I'm not convinced such a high level, short-format paper is appropriate for a new methodology. It's an odd setup to predicate the novelty on the methods, and then relegate all those methods to the end with a somewhat brief explanation, which makes it hard to meaningfully assess the results. But this is interesting work taking first step toward an important problem, which the introduction makes a strong case for.

I'm left wondering about the evidence that this is a better model than alternatives, as claimed or alluded to a few times. Sure, more parameters have been estimated from relatively few data points, but what is this a better model than? When does one need this complexity to model dynamics? Always? And where are the major ecological insights, as one would expect in this format? I'm sure plenty are possible, but they don't come out in the paper. For example, this seems to be a useful ecological insight (l 113):

'This suggests that shrubs ... were more resistant to temperature stress but had lower competitive ability than other species across the temperature gradient.'

But I'm still left wondering, 'so what'? It seems like there's a missing link these sorts of statements and the interesting big-picture context in the paragraph on l 173.

Could you explain the assumptions behind using the community abundance to inversely infer demographic parameters? For example, does it confound local recruitment and immigration? How is it feasible to pin down demographic parameters that could trade off, such as high survival and high recruitment vs low survival and low recruitment?

Relatedly, on 148, is an important point here that informative priors are critical for taming the many parameter tradeoffs possible in an inverse model?

158: Seems like a pretty bold claim. What about ecotypic variation in traits? Are you implicitly thinking about pretty small extents where this is minimal? What about demographic compensation (Villemas et al. 2015, Ecology) varying across a distribution and confounding these higher level demographic parameters that sort of integrate over them?

While the methods are clearer, they are still likely insufficient for most quantitative ecologists to get much from. Some context and examples illustrating the various steps with data could improve this.

Reviewer #2 (Remarks to the Author):

Review: NCOMMS-19-41011A

General:

I continue for this second version the praise that I expressed for the first version on the novelty of the paper. All issues that I found in the previous version, particularly about which demographic rate parameters can be estimated by inverse modelling, have been solved in the revision (if g is set explicitly to 1).

There are four main points that deserve attention and a number of details.

The four main points are

(1) The paper would be stronger/better/more scientific in my view if the limitations of the fitting of static data to an ODE process model using the equilibrium assumption would be voiced explicitly instead of implicitly as in the current version. This is not about the validity of the equilibrium assumption, but about what aspect of the parameters can and cannot be estimated from static data. The practical consequence of the limitation is that the calibrated model can be used to estimate the equilibria under new environmental conditions or new species with known trait

values. Absolute rates (and thus trajectories, i.e. explicit time dependent community assembly) cannot be predicted, I would say.

(2) In the paper, the parameters of the ODE process model are simply generalized linear functions of traits (generalized linear as in GLM). The regression coefficients of these functions (one for each demographic parameter) are mapped to a hypersphere with a location (mean), radius (standard deviation or norm) and angles with respect to the three synthetic trait axes which are principal components of the measured traits. The paper thus uses a hyper-spherical parametrization instead of using the usual Cartesian coordinate system. In the paper the parameters in the hyper-spherical parametrization are referred to as correlations. However, these parameters have in my view no interpretation as correlations in any usual sense of the word (they are a bit closer to partial correlations, as they are transformed regression coefficients, and any regression coefficient in a model with multiple predictors (traits) represents a partial or conditional effect and is thus a 'partial' regression coefficient). The term correlation coefficient should therefore be avoided. Call them simply angles? [Ok correlations can also be viewed as angles but this is where the analogy ends.]

(3) The paper mentions several advantages of the hyper-spherical parametrizing and I will not quibble about most of them; they may make sense. But, to me it makes no sense to state that they have the interpretation as correlation between trait and demographic parameter. Note that the traits in the fitting algorithm are principal components (and thus do not have a simple interpretation) and also that the main text does not report any of the angles/anything of the transformed space. The results in the main text are correlation between the measured traits and the calibrated demographic parameters. The angles and so on (i.e. the parametrization of the transferfunction) play no role in the main reporting.

(4) In line with the suggestion by (van Mourik et al. 2014) I would welcome as SI a data set with (at least) 1000 samples of the Bayesian algorithm of all 11 unknown parameters so that later researchers may investigate more properties of the model than those reported in the paper.

Details and remarks:

L10 Delete "only"

L29 can -> can be

L30 Delete "hereafter called demographic parameters" as this is repeated on line 46. But on L36 insert "of such models" after "parameters" and insert "rate" before parameters. Perhaps delete "demographic" here and on L43, where "rates" must then be "rate parameters".

L49 traits4?

L49-50 no sentence.

L53-55 Ok, this is perhaps not the place to indicate the limitations of the approach, but weaker words than "solve" and "enables" would be prudent in my view.

L56-60 These lines are not needed in the intro. The results reported in the main text do not require any of this detail. Move to "Methods". Moreover, alternative, more traditional parametrizations of the transfer function are equally valid, more easy to understand and probably even more easy to fit using the Bayesian approach of parameter estimation used in the paper.

L76 'four'? You mention three parameters thereafter. Please stay with four and add the missing one, in the first version of the paper described as: "within-species sensitivity of growth rate along the temperature gradient". (I suggest to describe this as second parameter as in the original submission, with deletion of "sensitivity of").

L77 Please add here [or with more rephrasing on line 80-81] something like "Because we use only static data, we can estimate only ratios of the last three parameters (Supporting Information 2.3). Therefore, we set the within-species growth rate to 1 for all species so that the last two parameters are ratios with respect to this rate." Note that g_i is dependent on the measurement scale of θ_j anyway.

L75-84 Somewhere you need to state that there is one parameter less you can fit because you chose to fit using the multinomial distribution (Supporting Information 2.3). If one can trust abundance in the community data as absolute amounts, you can fit using the Poisson or Tweedy distribution and this one parameter can be fit. [Cf fitting in multinomial regression where often the last category (here, the 118th species) is deleted from the data. E.g <https://stats.idre.ucla.edu/r/dae/multinomial-logistic-regression/#>, but here the situation is slightly different as shown in SI 2.3]

L88 "but it partly underestimated competitive exclusion (see Supplementary Fig. 5)." How did you reach this conclusion; on what is it based? Nor SI fig 5 nor SI 3.2 say anything about it.

L99 Thetamin and I have not been defined.

L97-127 What consequences do the above described aliases/non-identifiability's have for the interpretation?

L140 Discuss the limitation of fitting to the static data even if the equilibrium assumption holds true.

L141 The proposed transfer function is "based on". It does not rely on it (which sounds to me as requirement); other more traditional parametrizations could have been used without much change as explained in the Rebuttal by "Note also that our if we had specify [sic] a classical linear combination: $d = x_1 * t_1 + x_2 * t_2 + x_3 * t_3 + x_4$," and the words "[]way to compare a classical combination with our approach is to think about it as if we estimate the x_1 , x_2 and x_3 parameter while keeping the total $x_1^2 + x_2^2 + x_3^2$ value as a single parameter to be evaluated or kept constant if needed (like we did for the demographic parameter c)."

So instead, you could note that the parametrization is the hyperspherical parametrization which the multidimensional extension of the transformation going from cartesian to polar (2d) and spherical (3d) Please add this insight to SI. [search for a math book? All recent papers use it for correlation and partial correlation matrices, but that is out of order in the context of the paper, eg (Forrester & Zhang 2020)]. This insight makes section SI 3.4 redundant/trivial. See further main points (2) and (3) above.

L141-154 Move to SI. [This gives space for the remark at L140.]

L145 The parameters are not correlations in any usual sense. For a start, they depend on the scaling of predictors (here the traits, which are actually Principal components in the paper). Standardized regression coefficients would be scale-invariant. See further main points (2) and (3) above.

L248 The reader may wonder why the parameter specifying the temperature dependent growth is set constant across species. See main point (1).

L248 g is left unspecified later on. I guess it was set to 1. For generality, resort to g_i as in the first submission and discuss that g , c and l are defined only up to proportionality per species when fitting the equilibrium model to static data that one can set $g_i = 1$ for all i "to explore the same space", with as consequence that c and l are ratios wrt to g .

L276-281 I suggest to rephrase to something like: "For each of the three vectors of demographic parameters we formulated a transfer function, log-linear ones for c and l , as these must be positive and a linear one for θ_{\min} as this parameter can be both positive and negative. We used a hyperspherical parametrization of the regression coefficients, which has advantages over the usual cartesian parametrization as one can set priors for mean and variance of the regression coefficients (see SI). Hyperspherical coordinates are the multidimensional extension of polar coordinates in two dimensions. In detail, ... eq 2 and eq 3"

L281 Please avoid the word correlation as the parameters are not the usual correlations, not even partial correlation or standardized regression coefficients.

L288-302 I remark that, in terms of generalized linear models (GLM) that are familiar to statisticians and many biologists, Equation (3.1) defines a loglinear model for c and l and a linear model for θ_{\min} . So statisticians would call E the link scale.

L310-311 This extra non-identifiability is due to the use of the multinomial instead of the Poisson likelihood, but the multinomial is only mentioned at L317.

Figure 2 legend. As these demographic parameters are calibrated [and the correlations are not], move "Calibrated" to after "between" and "among".

Supporting info

Section 2.2

Please insert a reference to May's classic text "Stability and complexity in model ecosystems" which contains the essential of this section or say in which sense what you prove differs from this text.

Capital B is used in two meanings. Change notation.

matrice A?? Change notation to a more usual one, e.g. matrices (capitals) and vector (lower case) bold.

Section 2.3

"2.3 Why g is kept constant across species?" Change to "Why is g_j set to 1." Or something similar to describe the aliases in the system/the unidentifiability's in the system.

Section 2.4

The notation $b(c)$ and $b(l)$ should be explained.

There are too many errors to note all, but the results are correct in my view.

The x for matrix multiplication looks quite strange to me; please change to standard notation (i.e. delete x and be clear about a matrix transpose). The first displayed equation does not yield a column vector. If $B*_j$ is a column vector then the second displayed equation does not yield a scalar.

Section 3.1

Please remove the wording correlation matrix parametrization. It makes no sense to me. You simply define the regression coefficient (on the link scale) using the hyper-spherical coordinate system instead of the Cartesian one. This insight makes SI Fig 3 and Section 3.4 redundant. The ϕ are angles but not (partial) correlations in any known sense.

SI figure 4. What is "y". Clarify earlier that it is ϕ_1 in top row and ϕ_2 in bottom row. So, change the titles of the graphs, replacing y by ϕ_1 or ϕ_2 .

Cajo ter Braak Wageningen, 18-08-2020

Forrester, P.J. & Zhang, J. (2020) Parametrising correlation matrices. *Journal of Multivariate Analysis*, 178, 104619.<http://www.sciencedirect.com/science/article/pii/S0047259X19305330>
van Mourik, S., ter Braak, C., Stigter, H. & Molenaar, J. (2014) Prediction uncertainty assessment of a systems biology model requires a sample of the full probability distribution of its parameters. *PeerJ*, 2, e433.<http://dx.doi.org/10.7717/peerj.433>

REVIEWER COMMENTS

Reviewer #1 (Remarks to the Author):

Overall, I think this is a great set of ideas, but I'm not convinced such a high level, short-format paper is appropriate for a new methodology. It's an odd setup to predicate the novelty on the methods, and then relegate all those methods to the end with a somewhat brief explanation, which makes it hard to meaningfully assess the results. But this is interesting work taking first step toward an important problem, which the introduction makes a strong case for.

Following yours and the editor recommendation, we restructured the article. Now the community model is entirely presented in the first part of the results section and a summary of the transfer function is presented. The detailed description of the transfer function was kept for the method section.

I'm left wondering about the evidence that this is a better model than alternatives, as claimed or alluded to a few times. Sure, more parameters have been estimated from relatively few data points, but what is this a better model than? When does one need this complexity to model dynamics? Always?

Thank you for bringing this up, we agree that this is a crucial point, and we have addressed this question thoroughly in our revision. A detailed response and overview of changes is provided in response to the editor above, who highlighted the same question.

And where are the major ecological insights, as one would expect in this format? I'm sure plenty are possible, but they don't come out in the paper. For example, this seems to be a useful ecological insight (l 113):

'This suggests that shrubs ... were more resistant to temperature stress but had lower competitive ability than other species across the temperature gradient.'

But I'm still left wondering, 'so what'? It seems like there's a missing link these sorts of statements and the interesting big-picture context in the paragraph on l 173.

We agree that the calibrated relationships were not explicitly discussed in the discussion part. We now summarize them and link them to other work. We further improved the discussion to better link our results to the big-picture context.

l. 208: "An important point is that our approach does not rely on *a priori* assumptions about the demographic trade-offs and relevant trait-demography relationships, they rather emerge from the inverse modeling approach.

In our case study, we made the conjecture that species demography follows the stress-dominance hypothesis, and we ultimately quantified its support from the data and estimated the demographic trade-offs necessary to explain that theory. These corresponded to (i) a competition-stress tolerance trade-off that matched closely traits associated with the fast-slow leaf-economics trade-off and (ii) an axis of variation in intraspecific competition that matched traits related to plant size. These relationships are backed by knowledge about the global spectrum of plant trait variation and its relationship to plants' ecological niche and demography^{3,4,42}. While these calibrated functional trait-demography relationships are unsurprising, they are ecologically sensible and allow us to validate the calibration of the transfer function *a posteriori*. They further suggests that, when moving

towards more complex demographic models, a lack of unanimous knowledge about demography-trait relationships may not be an insurmountable obstacle.”

Could you explain the assumptions behind using the community abundance to inversely infer demographic parameters? For example, does it confound local recruitment and immigration? How is it feasible to pin down demographic parameters that could trade off, such as high survival and high recruitment vs low survival and low recruitment?

Based on your comment and Reviewer #2 comment about the equilibrium assumption, we now discuss the assumptions behind using static abundance data. Directly related to our case study, we now state how this led us to keep one demographic rate constant, thus “limiting our ability to fully model the interspecific variability of plant response to temperature and biotic stress” (1.254). For the sake of concision, we do not discuss explicitly local recruitment and immigration, but this is mentioned when we discuss “a vast corpus of models that study communities in terms of spatio-temporal dynamics, population structure, stability and alternative stable states^{49,50} with some approaches making explicit references to functional trait theory^{51,52}”. Citation #51 in particular (Falster et al. 2017), refers to a theoretical model that explicitly distinguishes plant recruitment, growth and survival in a functional trait framework.

Relatedly, on 148, is an important point here that informative priors are critical for taming the many parameter tradeoffs possible in an inverse model?

In our specific case, yes. While the phi parameters (angle parameters) that set the linkage between traits and demographic parameters were quick to converge, the other parameters (mean and variance) were more slow and prone to trade-off with each other. They also were important to control the ODE behaviour. This is now added in the detailed description of the transfer function.

I. 393 : “This hyperspherical parameterization is mathematically equivalent to a classical linear combination, albeit less intuitive. Compared to the latter, it allowed us to set regularizing priors⁶⁴ on the mean and variance of the demographic parameters (through parameters of a_m and b_m). This ensured the convergence of the ODE model (Supplementary Information 2.5) and controlled parameter trade-offs, but also allowed us to maintain uninformative priors on the parameters $\{\phi_{m,n}\}$ and avoid making prior assumptions on trait-demography relationships.”

158: Seems like a pretty bold claim. What about ecotypic variation in traits? Are you implicitly thinking about pretty small extents where this is minimal? What about demographic compensation (Villellas et al. 2015, Ecology) varying across a distribution and confounding these higher level demographic parameters that sort of integrate over them?

Our point here was not that intraspecific trait variability is unimportant. It is mostly to discuss the issue of parameters such as pairwise interaction coefficients that relate to two or more species. We amend the sentence to avoid that confusion.

I. 239: “The only requirement is that the process must be formulated as a function of attributes that can be linked to species-specific (or individual-specific) functional traits.”

While the methods are clearer, they are still likely insufficient for most quantitative ecologists to get much from. Some context and examples illustrating the various steps with data could improve this.

Due to the restructuring of the article towards a more methodological format, we hope that it is now clear. We further re-structure the part ‘Parameter Estimation’ (now “Bayesian Inference of the transfer

function parameters” l. 399) with sub-headers (Likelihood function; Priors; Posterior Estimation) to improve the readability of the methods.

Reviewer #2 (Remarks to the Author):

Review: NCOMMS-19-41011A

General:

I continue for this second version the praise that I expressed for the first version on the novelty of the paper. All issues that I found in the previous version, particularly about which demographic rate parameters can be estimated by inverse modelling, have been solved in the revision (if g is set explicitly to 1).

There are four main points that deserve attention and a number of details.

The four main points are

(1) The paper would be stronger/better/more scientific in my view if the limitations of the fitting of static data to an ODE process model using the equilibrium assumption would be voiced explicitly instead of implicitly as in the current version. This is not about the validity of the equilibrium assumption, but about what aspect of the parameters can and cannot be estimated from static data. The practical consequence of the limitation is that the calibrated model can be used to estimate the equilibria under new environmental conditions or new species with known trait values. Absolute rates (and thus trajectories, i.e. explicit time dependent community assembly) cannot be predicted, I would say.

We now state explicitly those limitations:

l. 411: “Because we calibrated our model using static relative abundances data and we used a likelihood function that follows a multinomial distribution, the demographic parameter g_i and the parameter b_c associated with the demographic parameter vector \mathbf{c} , were not identifiable (see Supplementary information). In consequence, they were fixed (for all species i , $g_i = 10^{-3.43}$ and $b_c = 10^{-3.8}$, Equation 1 and Equation 3). Those parameters would have been identifiable if the dataset included absolute abundance data (parameter b_c) or temporal data (demographic parameter g_i).”

And use that limitation for our discussion argument:

l. 252: “In our case study, assuming community equilibrium led us to fix one demographic rate across species (see methods), thus limiting our ability to fully model the interspecific variability of species’ responses to temperature and biotic stress.”

(2) In the paper, the parameters of the ODE process model are simply generalized linear functions of traits (generalized linear as in GLM). The regression coefficients of these functions (one for each demographic parameter) are mapped to a hypersphere with a location (mean), radius (standard deviation or norm) and angles with respect to the three synthetic trait axes which are principal components of the measured traits. The paper thus uses a hyper-spherical parametrization instead of using the usual Cartesian coordinate system. In the paper the parameters in the hyper-spherical

parametrization are referred to as correlations. However, these parameters have in my view no interpretation as correlations in any usual sense of the word (they are a bit closer to partial correlations, as they are transformed regression coefficients, and any regression coefficient in a model with multiple predictors (traits) represents a partial or conditional effect and is thus a ‘partial’

regression coefficient). The term correlation coefficient should therefore be avoided. Call them simply angles? [Ok correlations can also be viewed as angles but this is where the analogy ends.]

We removed the qualification of the phi parameters as correlation parameters and called them angles and added the hypersphere/polar coordinate system notions when necessary. We detailed later in response to your individual comments which parts of the manuscript and the supplementary materials were changed in that respect.

(3) The paper mentions several advantages of the hyper-spherical parametrizing and I will not quibble about most of them; they may make sense. But, to me it makes no sense to state that they have the interpretation as correlation between trait and demographic parameter. Note that the traits in the fitting algorithm are principal components (and thus do not have a simple interpretation) and also that the main text does not report any of the angles/anything of the transformed space. The results in the main text are correlation between the measured traits and the calibrated demographic parameters. The angles and so on (i.e. the parametrization of the transferfunction) play no role in the main reporting.

As pointed by the reviewer, the use of PCA axes does not lend itself well to an intuitive interpretation of the calibrated transfer function parameters. That is why we use correlations between empirical trait values and demographic parameters rather than discuss the posterior distribution of the transfer function parameters (that is nonetheless available in the supplementary materials). This is now laid out at the beginning of the result paragraph (it was previously stated in the methods):

l.158: “To study the output of the calibrated community model, we assessed the posterior correlation among the calibrated demographic parameters, the correlation between demographic parameters and observed functional traits and the distribution of calibrated demographic parameters among functional groups (forbs, grasses, legumes and shrubs). We rather related demographic parameters to the observed species functional traits values used to generate the PCA trait axes, rather than to the PCA trait axes themselves to facilitate the ecological interpretation of our results. The posterior distribution of the transfer function parameters is available in the Supplementary materials.”

Note that this paragraph was moved (and modified) from the methods in the previous version of the manuscript.

(4) In line with the suggestion by (van Mourik et al. 2014) I would welcome as SI a data set with (at least) 1000 samples of the Bayesian algorithm of all 11 unknown parameters so that later researchers may investigate more properties of the model than those reported in the paper.

Thanks, we agree that providing the MCMC chain is important. The entire chains are available in the code zip file submitted with the article (it will be uploaded as a Dryad repository) as Rdata files. We added a specific line to that in the supplementary materials. The code in the script output_analysis.R contains the necessary code line to access the whole posterior distribution and we added a comment line to explain what that code line does. We now further refer to this in the Supplementary materials. Furthermore, following the guidelines of Nature Communications, we have uploaded a Source Data folder that contains the raw data of each figures. We further added a csv file of posterior samples for the 11 parameters.

Supplementary materials 3.2: “The full posterior distribution of the parameter is stored alongside with the code to rerun the analysis.”

Details and remarks:

L10 Delete “only”

L29 can -> can be

Corrected.

L30 Delete “hereafter called demographic parameters” as this is repeated on line 46. But on

L36 insert “of such models” after “parameters” and insert “rate” before parameters. Perhaps delete “demographic” here and on l43, where “rates” must then be “rate parameters”.

Corrected.

L49 traits4?

It was a formatting error to refer to Adler et al. 2014, we fixed it.

L49-50 no sentence.

The sentence was shortened to “While the idea that form determines function is widely accepted, it would be extremely challenging to predict the nature of this relationship only from *a priori* assumptions. “

L53-55 Ok, this is perhaps not the place to indicate the limitations of the approach, but weaker words than “solve” and “enables” would be prudent in my view.

Replaced by “To tackle this challenge” and “allows” respectively.

L56-60 These lines are not needed in the intro. The results reported in the main text do not require any of this detail. Move to “Methods”. Moreover, alternative, more traditional parametrizations of the transfer function are equally valid, more easy to understand and probably even more easy to fit using the Bayesian approach of parameter estimation used in the paper.

That part was removed from the introduction.

L76 ‘four’? You mention three parameters thereafter. Please stay with four and add the missing one, in the first version of the paper described as: “within-species sensitivity of growth rate along the temperature gradient”. (I suggest to describe this as second parameter as in the original submission, with deletion of “sensitivity of”).

We named this parameter throughout the paper as “within-species variability of the growth rate along the temperature gradient”.

L77 Please add here [or with more rephrasing on line 80-81] something like “Because we use only static data, we can estimate only ratios of the last three parameters (Supporting Information 2.3). Therefore, we set the within-species growth rate to 1 for all species so that the last two parameters are ratios with respect to this rate.” Note that g_i is dependent on the measurement scale of θ_j anyway.

The three/four demographic parameters problem is now introduced in the introduction.

l. 83: “We formulated a community model derived from a Lotka-Volterra competition model that mimics these processes with four demographic parameters for each of the 118 species, three of which could be estimated with static community data.”

We do not name those parameters in the introduction though, they are now explained in detail in the first part of the results. The issue of identifiability for one of the demographic parameters is also mentioned in that part and better detailed in the methods and Supplementary information.

L75-84 Somewhere you need to state that there is one parameter less you can fit because you chose to fit using the multinomial distribution (Supporting Information 2.3). If one can trust abundance in the community data as absolute amounts, you can fit using the Poisson or Tweedy distribution and this one parameter can be fit. [Cf fitting in multinomial regression where often the last category (here, the 118th species) is deleted from the data. E.g <https://stats.idre.ucla.edu/r/dae/multinomial-logistic-regression/#>, but here the situation is slightly different as shown in SI 2.3]

This was stated and now expanded in the method parts. We further mention how temporal data and abundance data could solve those identifiability issues.

L 411: “Because we calibrated our model using static relative abundances data and we used a likelihood function that follows a multinomial distribution, the demographic parameter g_i and the parameter b_c associated with the demographic parameter vector c , were not identifiable (see Supplementary information). In consequence, they were fixed (for all species i , $g_i = 10^{-3.43}$ and $b_c = 10^{-3.8}$, Equation 1 and Equation 3). Those parameters would have been identifiable if the dataset included absolute abundance data (parameter b_c) or temporal data (demographic parameter g_i).”

L88 “but it partly underestimated competitive exclusion (see Supplementary Fig. 5).” How did you reach this conclusion; on what is it based? Nor SI fig 5 nor SI 3.2 say anything about it.

The reference was off. It was Supplementary Fig. 6 and the associated Supplementary materials 3.2. As we decided to remove that part of the Supplementary Materials, that statement was deleted.

L99 Thetamin and I have not been defined.

Abbreviations were added.

L97-127 What consequences do the above described aliases/non-identifiability’s have for the interpretation?

This is now stated in the discussion, as a part of a wider discussion on the case study assumptions.

L.252: “In our case study, assuming community equilibrium led us to fix one demographic rate across species (see methods), thus limiting our ability to fully model the interspecific variability of species’ responses to temperature and biotic stress.”

L140 Discuss the limitation of fitting to the static data even if the equilibrium assumption holds true.

We extended our discussion paragraph about temporal dynamic and community stability.

L. 248: “In our case study, we assumed that the observed plant communities were at equilibrium, a common assumption in spatial community modelling⁴⁸. Furthermore, we chose a globally-stable community model (Supplementary materials), which assumes that, for a given set of demographic parameters values and environmental conditions, a unique stable community structure will emerge.

These assumptions make the analysis more practical, but they may not always hold in reality^{12,48}. In our case study, assuming community equilibrium led us to fix one demographic rate across species (see methods), thus limiting our ability to fully model the interspecific variability of species’ responses to temperature and biotic stress. While this is arguably reasonable when modelling a relatively small static dataset without temporal data, it would be less adequate to model larger

spatio-temporal datasets where it would be more essential, and feasible, to distinguish and calibrate detailed demographic processes.

Theoretical ecology has produced a vast corpus of models that study communities in terms of spatio-temporal dynamics, population structure, stability and alternative stable states^{49,50}, with some approaches making explicit references to functional trait theory^{51,52}. This suggests that with adequate community models, our transfer function approach could be used to move away from the equilibrium assumption, and model spatial and temporal variation of species-rich communities. Besides the important data requirements, however, these more complex models would create new analytical challenges as simulation approaches (e.g. ODEs) might not be appropriate to efficiently characterize community dynamics and equilibrium states⁵³. In this context of increasing model and data complexity, assuming *a priori* that species demographic parameters depend on a limited number of functional traits will likely be a critical asset to study the model behaviors and reduce the complexity of its calibration at the onset.”

L141 The proposed transfer function is “based on”. It does not rely on it (which sounds to me as requirement); other more traditional parametrizations could have been used without much change as explained in the Rebuttal by “Note also that our if we had specify [sic] a classical linear combination: $d = x_1 * t_1 + x_2 * t_2 + x_3 * t_3 + x_4$,” and the words “[]way to compare a classical combination with our approach is to think about it as if we estimate the x_1 , x_2 and x_3 parameter while keeping the total $x_1^2 + x_2^2 + x_3^2$ value as a single parameter to be evaluated or kept constant if needed (like we did for the demographic parameter c).”

So instead, you could note that the parametrization is the hyperspherical parametrization which the multidimensional extension of the transformation going from cartesian to polar (2d) and spherical (3d). Please add this insight to SI. [search for a math book? All recent papers use it for correlation and partial correlation matrices, but that is out of order in the context of the paper, eg (Forrester & Zhang 2020)]. This insight makes section SI 3.4 redundant/trivial. See further main points (2) and (3) above.

This insight was added in the methods. We further refer to a classic reference about hyperspherical transformation of cartesian coefficients (Blumenson, L. E. A Derivation of n-Dimensional Spherical Coordinates. *Am. Math. Mon.* **67**, 63–66 (1960)).

I. 368: “The regression coefficients of the linear expressions linking the species demographic parameters values $d_{i,m}$ to the functional trait values $t_{i,n}$ can be viewed as coordinates lying on a unit hypersphere of dimension N. We express them using the multidimensional extension of the transformation of Cartesian coordinates to polar coordinates.”

L141-154 Move to SI. [This gives space for the remark at L140.]

The paragraph was deleted. A much shorter version was added in the method part after explaining the formulation of the transfer function.

I. 393: “This hyperspherical parameterization is mathematically equivalent to a classical linear combination, albeit less intuitive. Compared to the latter, it allowed us to set regularizing priors⁶⁸ on the mean and variance of the demographic parameters (through parameters of a_m and b_m). This ensured the convergence of the ODE model (Supplementary Information 2.5) and controlled parameter trade-offs, but also allowed us to maintain uninformative priors on the parameters $\{\phi_{m,n}\}$ and avoid making prior assumptions on trait-demography relationships.”

L145 The parameters are not correlations in any usual sense. For a start, they depend on the scaling of predictors (here the traits, which are actually Principal components in the paper). Standardized regression coefficients would be scale-invariant. See further main points (2) and (3) above.

We don't call them correlation parameters anymore.

L248 The reader may wonder why the parameter specifying the temperature dependent growth is set constant across species. See main point (1).

L248 g is left unspecified later on. I guess it was set to 1. For generality, resort to g_i as in the first submission and discuss that g , c and l are defined only up to proportionality per species when fitting the equilibrium model to static data that one can set $g_i = 1$ for all i "to explore the same space", with as consequence that c and l are ratios wrt to g .

We put back the description of g_i in that paragraph (now moved in the results). And specify later (l. 411) that its value was fixed across species because it is not identifiable.

l. 137: "In the absence of temporal data about our studied plant communities, we assumed that they were stable and at equilibrium and could be modeled from the ODE equilibrium. Because of this, within-species variability of the growth rate along the temperature gradient could not be estimated and was thus kept constant across species (see Methods and Supplementary information 2.3)."

L276-281 I suggest to rephrase to something like: "For each of the three vectors of demographic parameters we formulated a transfer function, log-linear ones for c and l , as these must be positive and a linear one for θ_{\min} as this parameter can be both positive and negative. We used a hyperspherical parametrization of the regression coefficients, which has advantages over the usual cartesian parametrization as one can set priors for mean and variance of the regression coefficients (see SI). Hyperspherical coordinates are the multidimensional extension of polar coordinates in two dimensions. In detail, ... eq 2 and eq 3"

We added some of your suggestions in this paragraph. It now reads as follows:

l. 360: "In our case, we used a linear function of the functional traits to specify θ_{\min} as it can take both positive and negative values across species and a log-linear function for l and c , that can take only positive values.

We used a hyperspherical parameterization of the regression coefficients of the linear and log-linear functions^{64,65}. That formulation defines the relationship between demographic parameters and functional traits with two sets of parameters: a first set that controlled the link between demographic parameters and functional traits, and a second set that controlled the mean and standard deviation of demographic parameters across species."

We added in the paragraph explaining the angle parameters ϕ_i : "The regression coefficients of the linear expressions linking the species demographic parameters values $d_{i,m}$ to the functional trait values $t_{i,n}$ can be viewed as coordinates lying on a unit hypersphere of dimension N . We express them using the multidimensional extension of the transformation of Cartesian coordinates to polar coordinates."

L281 Please avoid the word correlation as the parameters are not the usual correlations, not even partial correlation or standardized regression coefficients.

Correlation has been deleted throughout that paragraph except in one sentence that reads as follows:

I. 375: “This parameterization samples efficiently all possible correlations between each demographic parameter and functional trait while keeping constant the mean and standard deviation of the demographic parameters (Supplementary materials 3.1).”

L288-302 I remark that, in terms of generalized linear models (GLM) that are familiar to statisticians and many biologists, Equation (3.1) defines a loglinear model for c and l and a linear model for Θ_{\min} . So statisticians would call E the link scale.

We now call E the link scale.

L310-311 This extra non-identifiability is due to the use of the multinomial instead of the Poisson likelihood, but the multinomial is only mentioned at L317.

We moved the paragraph about identifiability after this paragraph.

Figure 2 legend. As these demographic parameters are calibrated [and the correlations are not], move “Calibrated” to after “between” and “among”.

Corrected

Supporting info

Section 2.2

Please insert a reference to May’s classic text “Stability and complexity in model ecosystems” which contains the essential of this section or say in which sense what you prove differs from this text.

We now introduced that part with a reference to May’s work.

Capital B is used in two meanings. Change notation.

Replaced by J

matrice A ? Change notation to a more usual one, e.g. matrices (capitals) and vector (lower case) bold.

Section 2.3

“2.3 Why g is kept constant across species?” Change to “Why is g_j set to 1.” Or something similar to describe the aliases in the system/the unidentifiability’s in the system.

Changed to “Why is g_i set constant?”

Section 2.4

The notation $b(c)$ and $b(l)$ should be explained.

It is an orphan notation formatting from the previous version of the manuscript. It is now changed to b_c and b_l . We added a new line to explain better those notations in relation to the transfer function equations.

There are too many errors to note all, but the results are correct in my view.

The x for matrix multiplication looks quite strange to me; please change to standard notation (i.e. delete x and be clear about a matrix transpose). The first displayed equation does not yield a column vector. If $B*_j$ is a column vector then the second displayed equation does not yield a scalar.

We corrected the transpose errors and the matrix notation.

Section 3.1

Please remove the wording correlation matrix parametrization. It makes no sense to me. You simply define the regression coefficient (on the link scale) using the hyper-spherical coordinate system instead of the Cartesian one. This insight makes SI Fig 3 and Section 3.4 redundant.

The phi are angles but not (partial) correlations in any known sense.

The title of the section is now “Properties of the hyperspherical parameterization” and in the text, we now say: “Here we illustrate that the distribution of the angle parameters $\{\phi_{i,n}\}$ allows an efficient sampling of all possible correlations between the vector $D=\{\text{dim}\}$ and the trait vectors contained in the matrix $T=\{t_{kj}\}$, for the case where $N = 3$.” instead of saying that the phi parameters can be assimilated to correlation parameters.

The legend of Supplementary Figure 3 is now “(partial) correlations between a response variable y and three trait vectors t_1 , t_2 and t_3 as a function of angle parameters ϕ_1 and ϕ_2 . The figure shows how the angle parameters allow the sampling of the correlation structure between y and t_1 , t_2 and t_3 .”

SI figure 4. What is “ y ”. Clarify earlier that it is ϕ_1 in top row and ϕ_2 in bottom row. So, change the titles of the graphs, replacing y by ϕ_1 or ϕ_2 .

Those graphs were misunderstood. As described in the legend, the two rows correspond to two different scheme of sampling ϕ_1 . Furthermore the graphics do display correlation coefficients values and not the distribution of angle parameters ϕ_1 and ϕ_2 . “ y ” is now described in the figure legend as “the demographic parameter vector y ” and “ y was calculated as a function of t_1 , t_2 and t_3 as specified by Equation 1.

Cajo ter Braak Wageningen, 18-08-2020

Forrester, P.J. & Zhang, J. (2020) Parametrising correlation matrices. *Journal of Multivariate Analysis*, 178, 104619. <http://www.sciencedirect.com/science/article/pii/S0047259X19305330>

van Mourik, S., ter Braak, C., Stigter, H. & Molenaar, J. (2014) Prediction uncertainty assessment of a systems biology model requires a sample of the full probability distribution of its parameters. *PeerJ*, 2, e433. <http://dx.doi.org/10.7717/peerj.433>

Reviewer comments, third round:

Reviewer #2 (Remarks to the Author):

Review: NCOMMS-19-41011B

General:

This is my third look at the paper. The authors took care of all my concerns and included the insights that I provided in a superb way. A few minor points to consider are as follows.

Details and remarks:

L440. Please reconsider "novelty"; rephrase the sentence. The method is novel anyway. Perhaps something on "the novel type of information that the new method generates"?

L276. [40] Laughlin et al 2012 is on models that use intraspecific trait variation and is a very complicated model consisting of several steps that are open for discussion; its behaviour is not well researched. Simpler robust models for species-specific trait data (and open for individually based trait-data) are in (sorry!) ter Braak (2019) and Jamil et al (2013).

Supporting info

Page 18 "our tested community model further seeks to model species hierarchy within communities, a feature absent from the fourth corner analysis." I did not understand the species hierarchy within communities. The name does not occur in the main text. Please explain and rephrase.

The reference for ade4 is given as Dray & Dufour 2007. A more modern reference is (Thioulouse et al. 2018)

Cajo ter Braak Wageningen, 20-01-2021

Jamil, T., Ozinga, W.A., Kleyer, M. & ter Braak, C.J.F. (2013) Selecting traits that explain species–environment relationships: a generalized linear mixed model approach. *Journal of Vegetation Science*, 24, 988–1000.<http://dx.doi.org/10.1111/j.1654-1103.2012.12036.x>

ter Braak, C.J.F. (2019) New robust weighted averaging- and model-based methods for assessing trait–environment relationships. *Methods in Ecology and Evolution*, 10, 1962–1971.<https://doi.org/10.1111/2041-210X.13278>

Thioulouse, J., Dray, S., Dufour, A.-B., Siberchicot, A., Jombart, T. & Pavoine, S. (2018) *Multivariate Analysis of Ecological Data with ade4*. Springer New York, New York, NY.978-1-4939-8850-1

Review: NCOMMS-19-41011B

General:

This is my third look at the paper. The authors took care of all my concerns and included the insights that I provided in a superb way. A few minor points to consider are as follows.

Thanks!

Details and remarks:

L440. Please reconsider “novelty”; rephrase the sentence. The method is novel anyway. Perhaps something on “the novel type of information that the new method generates”?

We replaced “novelty” by “usefulness”

L276. [40] Laughlin et al 2012 is on models that use intraspecific trait variation and is a very complicated model consisting of several steps that are open for discussion; its behaviour is not well researched. Simpler robust models for species-specific trait data (and open for individually based trait-data) are in (sorry!) ter Braak (2019) and Jamil et al (2013).

We now cite these two papers instead of Laughlin et al. 2012

Supporting info

Page 18 “our tested community model further seeks to model species hierarchy within communities, a feature absent from the fourth corner analysis.” I did not understand the species hierarchy within communities. The name does not occur in the main text. Please explain and rephrase.

We removed that statement. It was inadequate.

The reference for ade4 is given as Dray & Dufour 2007. A more modern reference is (Thioulouse et al. 2018)

Replaced